# Mechanical force promotes dimethylarginine dimethylaminohydrolase 1-mediated hydrolysis of the metabolite asymmetric dimethylarginine to enhance bone formation

Ziang Xie[1,2,6], Lei Hou[3,6], Shuying Shen[1,2,6], Yizheng Wu[1,2], Jian Wang[4], Zhiwei Jie[1,2], Xiangde Zhao[1,2], Xiang Li[1,2], Xuyang Zhang[1,2], Junxin Chen[1,2], Wenbin Xu[1,2], Lei Ning[1,2], Qingliang Ma[1,2], Shiyu Wang[1,2], Haoming Wang[1,2], Putao Yuan[1,2], Xiangqian Fang[1,2✉], An Qin[5✉] & Shunwu Fan[1,2✉]

Mechanical force is critical for the development and remodeling of bone. Here we report that mechanical force regulates the production of the metabolite asymmetric dimethylarginine (ADMA) via regulating the hydrolytic enzyme dimethylarginine dimethylaminohydrolase 1 (*Ddah1*) expression in osteoblasts. The presence of -394 4 N del/ins polymorphism of *Ddah1* and higher serum ADMA concentration are negatively associated with bone mineral density. Global or osteoblast-specific deletion of *Ddah1* leads to increased ADMA level but reduced bone formation. Further molecular study unveils that mechanical stimulation enhances TAZ/SMAD4-induced *Ddah1* transcription. Deletion of *Ddah1* in osteoblast-lineage cells fails to respond to mechanical stimulus-associated bone formation. Taken together, the study reveals mechanical force is capable of down-regulating ADMA to enhance bone formation.

[1] Department of Orthopedic Surgery, Sir Run Run Shaw Hospital, Zhejiang University School of Medicine, Hangzhou, China. [2] Key Laboratory of Musculoskeletal System Degeneration and Regeneration Translational Research of Zhejiang Province, Hangzhou, China. [3] Department of Cardiology, Shanghai Tongren Hospital, Shanghai Jiaotong University School of Medicine, Shanghai, China. [4] Department of Orthopaedics, Tongde Hospital of Zhejiang Province, Hangzhou, China. [5] Department of Orthopaedics, Shanghai Key Laboratory of Orthopaedic Implant, Shanghai Ninth People's Hospital, Shanghai Jiaotong University School of Medicine, Shanghai, China. [6] These authors contributed equally: Ziang Xie, Lei Hou, Shuying Shen. ✉email: orthofxq@zju.edu.cn; dr_qinan@163.com; shunwu_fan@zju.edu.cn

During the post-translational methylation of arginine residues within proteins and proteolysis, asymmetric demethylation (ADMA) is released into the cytoplasm[1]. ADMA has been generally implicated as an important risk factor for atherosclerosis, cardiovascular diseases and renal diseases[2–4]. ADMA is regarded as a competitive inhibitor of nitric oxide synthase (NOS) enzymes. Elevation of ADMA accelerates oxidative stress but reduces the production of NO[5,6]. It is suggested that ADMA remains stable until it is hydrolyzed to D-citrulline and dimethylamines by the hydrolytic enzyme, dimethylarginine dimethylamine hydrolase (DDAH)[7]. To date, two distinct DDAH isoforms (DDAH1 and DDAH2) have been identified[8]. DDAH1 has been confirmed to mainly contribute to the overall DDAH activity in several tissues[9]. Emerging evidences suggest that plasma ADMA levels are significantly associated with SNPs (Single Nucleotide Polymorphisms) in *Ddah1*, which contributed to cardiovascular diseases and diabetes[10,11]. A loss-of-function *Ddah1* promoter polymorphism is associated with increased susceptibility to cardiovascular diseases and thrombosis stroke[12]. Thus, these SNPs of *Ddah1* are likely associated with metabolic syndrome including metabolic bone diseases[13,14]. Therefore, our initial aim was to investigate if the loss-of-function *Ddah1* promoter polymorphism is associated with osteoporosis.

Previous studies suggested that deletion of NOS enzymes led to reduced bone formation, decreased osteoblast number and mineralization rates[15,16]. Additionally, it was suggested that NO regulated by NOS enzymes is responsive to mechanical force, which was important for bone homeostasis[17–20]. To note, ADMA is a competitive inhibitor of NOS enzymes[6], therefore, our second aim was to investigate whether ADMA hydrolyzed by DDAH is involved in bone remodeling responsive to mechanical force. Here, we found that associations between ADMA level, the -394 4N del/ins polymorphism of *Ddah1*, and bone mineral density were found in the Chinese population, suggesting the potential role of DDAH1 expression in bone health. We further explored the underlying mechanisms. Deletion of *Ddah1* in osteoblasts led to increased ADMA in serum and bone samples, which finally suppressed osteoblast differentiation and thus reduced bone mass in mice. Subsequently, we intended to explore which signaling pathway was involved in the regulation of DDAH1 mediated by mechanical force. Previous studies provided evidence that expression of DDAH1 was controlled by Yap1/Taz, also knocking down of Lats2 reduced the expression of DDAH1[21]. Not surprisingly, Yap1/Taz signaling pathway activated by mechanical force was confirmed by several groups, which implied that mechanical force might regulate DDAH1 expression via Yap1/Taz pathway[22–24]. The exercise was generally considered as an enhancer of mechanical stimulation to enhance bone formation[25]. Conversely, tail-suspension deprived the mechanical stimulation of bone that reduced the activation of mechanical-response signaling pathway[26]. Thus, we next performed these two models for exploring whether DDAH1/ADMA was regulated by mechanical force in vivo. Additionally, we found that deletion of *Ddah1* abolished the responsive ability of osteoblasts to mechanical force. Finally, treatment with the Ddah1 inhibitor (PD 404182) was unable to reduce the bone volume of *Ddah1* global knockout mice, however, administration of adeno-associated virus (AAV)-DDAH1 promoted bone healing in *Ddah1*[Prx1] conditional-knockout mice.

In brief, our data suggest that DDAH1/ADMA pathway response to mechanical force to regulate bone formation. The findings partially broadened the acknowledge of both mechanobiology and bone biology. Ddah1-targeted new compounds could be potential therapeutic approaches for treating bone-related diseases.

## Results

**The -394 4N del/ins polymorphism of *Ddah1* and increased level of ADMA is negatively associated with bone mineral density.** Given that loss-of-function polymorphism of *Ddah1* promoter was associated with increased susceptibility to metabolic syndrome (MS)[13], we asked whether ADMA level and *Ddah1* promoter polymorphism were associated with the bone mineral density (BMD) in humans. A Chinese population of 1404 participants were included, and 570 participants from the cohort were subjected to analyze ADMA concentrations. The characteristics of the participants are presented in Supplementary Table S1 and Supplementary Table S2. A novel -394 4N del/ins polymorphism of *Ddah1* promoter region was previously reported to affect the mRNA levels of *Ddah1* in patients[12]. Consistent with the previous study, the mRNA level of *Ddah1* was lower in participants with the del/ins and ins/ins polymorphisms (Fig. 1a). The median serum ADMA concentrations were significantly higher in individuals with one or both copies of the -394 4N ins allele ($206.3 \pm 26.2$ ng/ml or $181.1 \pm 8.6$ ng/ml, respectively) compared to individuals with both copies of the -394 4N del allele

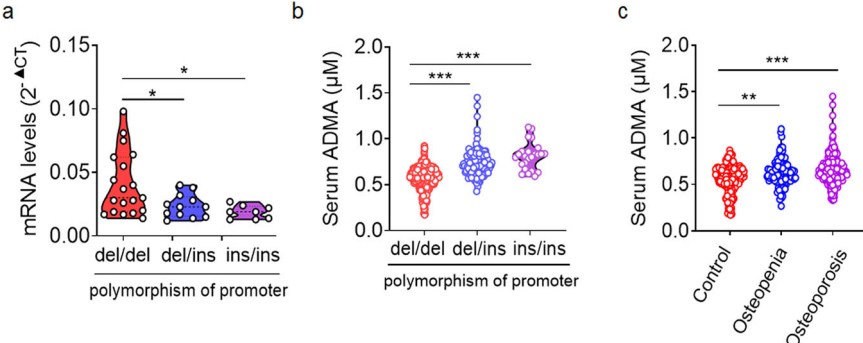

**Fig. 1 Association of ADMA levels and the -394 4N del/ins polymorphism of *Ddah1* with bone mineral density in a large Chinese population-based study. a** qPCR of Ddah1 mRNA in hemocytes from patients with -394 del/del, -394 del/ins and -394 ins/ins Ddah1 genotypes. **$p < 0.01$. del/del, $n = 19$. del/ins, $n = 12$. ins/ins, $n = 8$. Data are represented as mean values ± SD. The data were analyzed by an unpaired two-tailed Student's *t* test in all panels. **b** Results from LC-MS assay for ADMA concentrations of the serum samples from -394 del/del, -394 del/ins and -394 ins/ins patients. **$p < 0.01$. del/del, $n = 420$. del/ins, $n = 126$. ins/ins, $n = 24$. Data are represented as mean values ± SD. The data were analyzed by an unpaired two-tailed Student's *t* test in all panels. **c** Results from LC-MS assay for ADMA concentrations of the serum samples from control, osteopenia and osteoporosis participants. **$p < 0.01$. ***$p < 0.005$. control, $n = 190$. osteopenia, $n = 117$. osteoporosis, $n = 263$. Data are represented as mean values ± SD. The data were analyzed by an unpaired two-tailed Student's *t* test in all panels.

**Table 1 Association between the -394 4N del/ins polymorphism of Ddah1 and osteoporosis, analyzed by logistic regression.**

| Samples (n = 1404) | Genotype, n (%) | | | 4N del/ins + ins/ins | | |
| --- | --- | --- | --- | --- | --- | --- |
| | del/del | del/ins | ins/ins | Frequency % | Unadjusted OR (95% CI) | Adjusted OR (95% CI) |
| Control (n = 470) | 378 (80.4) | 84 (17.9) | 8 (1.7) | 19.6 | Reference | |
| Osteopenia (n = 635) | 464 (73.1) | 151 (23.8) | 20 (3.1) | 26.9 | 1.45 (1.13-1.86)[&] | 1.42 (1.07-1.88)[*] |
| Osteporosis (n = 299) | 219 (73.2) | 70 (23.4) | 10 (3.4) | 26.8 | 1.45 (1.08-1.94)[*] | 1.57 (1.03-2.40)[*] |

Odds ratios (ORs) and 95% confidence intervals (CIs) were obtained by logistic regression, with and without adjustment for sex, age, and body mass index.
[*]$p < 0.05$. [&]$p < 0.01$. OR, an indicator of the degree of association between exposure and disease, and was analyzed by logistic regression.

(158.1 ± 3.0 ng/ml; Fig. 1b). Furthermore, the plasma ADMA concentrations in 570 participants with normal BMD, osteopenia, or osteoporosis was examined. The serum ADMA concentrations were significantly higher in individuals with osteopenia and osteoporosis than in control subjects (Fig. 1c). The association of the -394 4N del/ins polymorphism with osteoporosis was also investigated based on 1404 participants. One or both copies of the -394 4N ins allele was significantly associated with increased risks of developing osteopenia and osteoporosis, both with or without adjustment for conventional risks, including age, sex, and the body mass index (Table 1). Compared with the control group, the group with osteopenia had an unadjusted odds ratio (OR) of 1.45 ($P = 0.004$) and an adjusted OR of 1.42 ($P = 0.015$). The group with osteoporosis had an unadjusted OR of 1.45 ($P = 0.015$) and an adjusted OR of 1.57 ($P = 0.037$). Taken together, these results provide evidence that the -394 4N del/ins polymorphism of *Ddah1* was associated with bone health, as well as the serum concentration of ADMA.

**Ddah1 deficiency but not Ddah2 resulted in bone loss and weakened bone formation.** According to the loss-of-function polymorphism of *Ddah1* was related to BMD in the population, we intended to study the effects of DDAH1 on bone metabolism. In this study, *Ddah1* global knockout mice was used[27]. As shown, 10-week old *Ddah1*$^{-/-}$ mice displayed decreased BV/TV ratios in the femur and also a decrease in the cortical bone thickness, compared with wild-type (WT) control mice (Fig. 2a, b). Consistently, the histomorphometric analysis showed that the MAR, BFR/BS and Ob.S/BS were all reduced in *Ddah1*$^{-/-}$ mice compared with WT mice (Fig. 2c, d). Furthermore, the results of the ELISA assay were demonstrated that *Ddah1* deficiency led to a decrease of P1NP concentration in serum (Fig. 2e). Additionally, immuno-fluorescence staining of bone tissues showed that *Ddah1*$^{-/-}$ mice had fewer osteocalcin-positive (OCN$^+$) osteoblasts, compared to *Ddah1*$^{+/+}$ mice (Fig. 2f). Interestingly, knocking out of *Ddah1* did not affect osteoclast number but slightly increased osteoclast surface of bone surface (Oc.S/BS) in *Ddah1*$^{-/-}$ mice (Fig. 2g, h). Finally, we also generated *Ddah2*$^{-/-}$ mice and found global knockout of *Ddah2* had no effects on bone mass (Fig. 2i, j), which might be due to unchanged concentrations of ADMA (Supplementary Fig. 1). Taken together, these data supported that deletion of *Ddah1* led to bone loss and weakened bone formation.

**Ddah1 deficiency in mesenchymal stem cells led to bone loss and reduced bone mechanical properties.** Given that global knockout of *Ddah1* led to bone loss and dramatically decreased bone formation, we investigated whether deletion of *Ddah1* in osteoblast-lineage cells regulates bone formation. Firstly, we analyzed the mRNA level and protein level in osteoblasts, and the results demonstrated that DDAH1 expression was upregulated in osteogenesis (Fig. 3a–d). Next, we crossed *Ddah1*$^{f/f}$ mice with Prx1-Cre mice to generate *Ddah1*$^{Prx1}$ conditional-knockout

(cKO) mice. The *Ddah1*$^{Prx1}$ cKO mice appeared normal at birth and later in life, and DDAH1 protein expression was barely detectable in BMSCs from *Ddah1*$^{Prx1}$ mice (Supplementary Fig. 2). Micro-computed tomography (CT) analysis of the femur metaphysis of 12-week old mice showed significantly decreased bone BV/TV and Tb.N in *Ddah1*$^{Prx1}$ cKO mice relative to their *Ddah1*$^{f/f}$ littermates, whereas no significant alterations of these bone parameters were observed in 4-week old *Ddah1*$^{Prx1}$ cKO mice (Fig. 3e, f). Furthermore, 12-week-old *Ddah1*$^{Prx1}$ mice had decreased BFR/BS and MAR values, when compared with the control mice (Fig. 3g, h). In addition, the osteocalcin-positive (OCN$^+$) osteoblasts of 12-week-old *Ddah1*$^{Prx1}$ cKO mice were also dramatically reduced compared to those of *Ddah1*$^{f/f}$ mice (Fig. 3i, j). Consistent with these data, deletion of *Ddah1* in MSCs also impaired the mechanical properties of bones in *Ddah1*$^{Prx1}$ mice (Fig. 3k). In general, these results supported that *Ddah1* deficiency in osteoblast-lineage cells contributed to bone loss and reduced bone mechanical properties by impairing bone formation.

**Loss of Ddah1 accumulated ADMA to induce bone loss in vivo.** To explore how deletion of *Ddah1* led to bone loss in vivo, we first intended to analyze whether ADMA was involved in bone loss in vivo. Thus, we analyzed the concentrations of ADMA in serum and bone tissues, and we found that concentrations of ADMA were both dramatically increased in *Ddah1*$^{-/-}$ and *Ddah1*$^{Prx1}$ mice compared with the littermate *Ddah1*$^{+/+}$ and *Ddah1*$^{f/f}$ mice, respectively (Fig. 4a, b). Simultaneously, we performed an immunofluorescence assay to analyze the level of endothelial nitric oxide synthase (eNOS) in tibiae (Fig. 4c, d). The data showed that deletion of DDAH1 in osteoblasts caused elevated ADMA, which then caused impaired NO production in tibiae. To further confirm that accumulating ADMA led to bone loss and reduced bone formation, we bred mice with a high dose of ADMA, and serum concentrations of ADMA suggested that ADMA was accumulated in vivo (Fig. 4g). Indeed, mice treated with ADMA had decreased trabecular bone volume, trabecular bone number, and cortical bone thickness (Fig. 4e, f). While ADMA-treated mice also had decreased concentrations of P1NP in serum (Fig. 4h). Consistent with the previous data, ADMA-treated mice showed a weakened bone formation compared with vehicle group mice (Fig. 4i, j). As far as mechanical properties are concerned, treatment with ADMA impaired stiffness and the ultimate force of femurs compared with control group mice (Fig. 4k). Since knockout of *Ddah1* resulted in impaired bone formation, we concluded that deletion of *Ddah1* led to accumulated ADMA to inhibit NO production and thus impaired bone formation in mice.

**The expression of DDAH1 was a response to mechanical force via TAZ/SMAD4 signaling pathway.** NO synthesis is regulated by mechanical force, which is critical for the skeletal system[17–19].

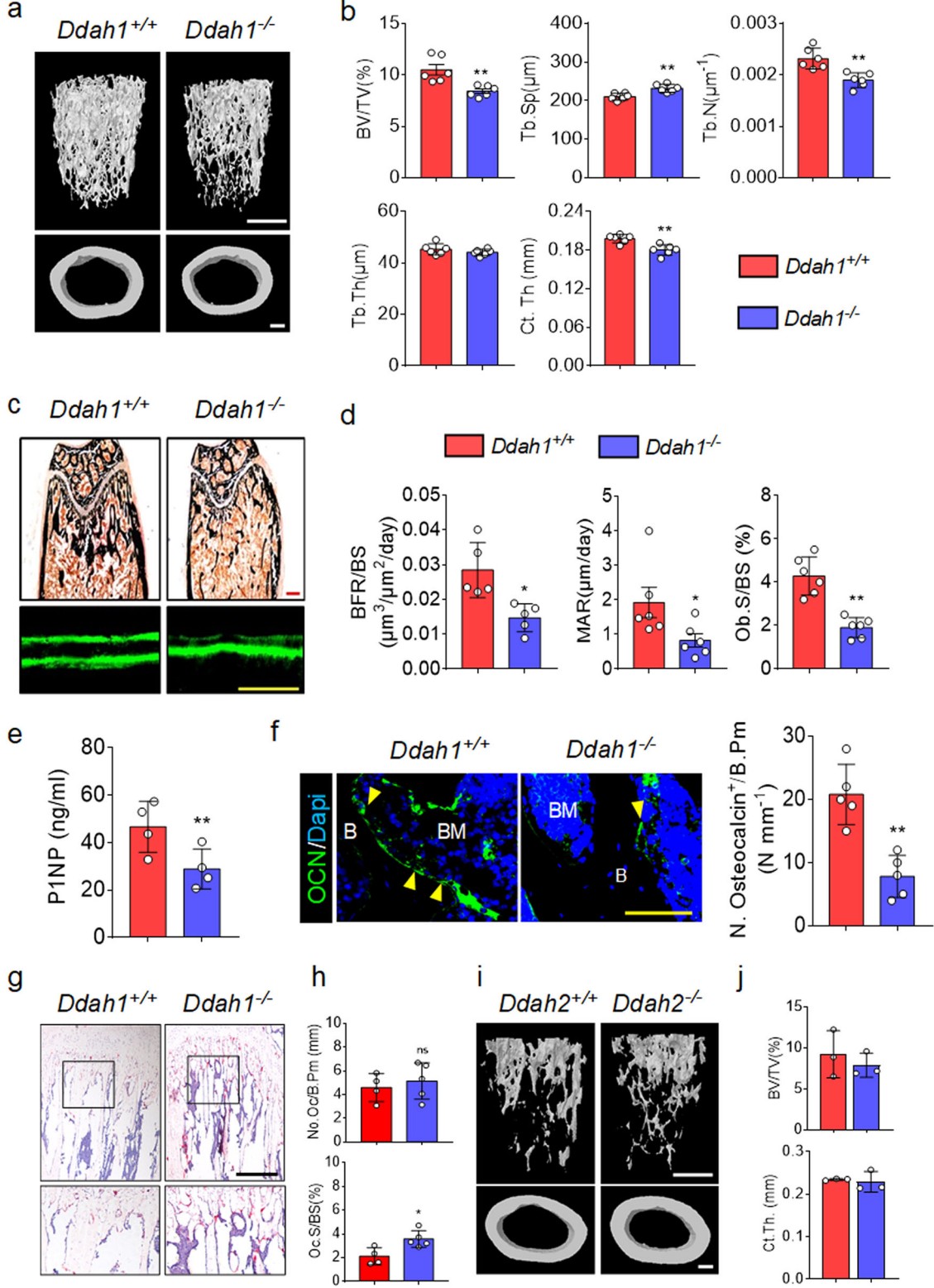

Therefore, we proposed whether DDAH1 mediated ADMA hydrolysis to link mechanical force and NO synthesis. In an attempt to address the hypothesis, we firstly analyzed the expression of DDAH1 and DDAH2 in unloading and exercise mice. Intriguingly, we found that DDAH1 was increased in the bones of exercise mice but was decreased in the bones of unloading mice (Fig. 5a, b). Simultaneously, we performed metabonomics to explore the changed metabolites responsive to

mechanical force. The data showed that ADMA was significantly decreased in serum samples of mice after exercise treatment (Fig. 5c, d). Further LC-MS assay confirmed that the concentration of ADMA was reduced in serum samples of exercise mice, but it was increased in serum samples of unloading mice (Fig. 5e). Interestingly, although the concentrations of ADMA and arginine were upregulated in exercise treatment mice, NO production was increased in the exercise mice (Supplementary Fig. 3). Next, we

**Fig. 2 *Ddah1* deficiency induces bone loss by impairing bone formation in vivo. a** Representative micro-CT images of trabecular bone and cortical bone in the distal femur (left). Scale bars = 200 μm. **b** Bone volume/total volume (BV/TV), trabecular bone thickness (Tb.Th), trabecular bone number (Tb.N), trabecular bone separation (Tb.Sp), and cortical bone thickness (Ct.Th) in WT mice (*Ddah1*$^{+/+}$, n = 6) and Ddah1 knock out mice (*Ddah1*$^{-/-}$, n = 6) male mice at 10 weeks of age. **p < 0.01. Data are represented as mean values ± SD. **c** Representative histomorphometric images of the femur and calcein double staining. Scale bars, 100 μm. **d** Quantification of histomorphometric parameters of femur in *Ddah1*$^{+/+}$ and *Ddah1*$^{-/-}$ mice at 10 weeks of age. *Ddah1*$^{+/+}$, n = 6. *Ddah1*$^{-/-}$, n = 6. *p < 0.05. **p < 0.01. Data are represented as mean values ± SD. **e** Concentrations of P1NP from the serum of *Ddah1*$^{+/+}$ and *Ddah1*$^{-/-}$ mice. *Ddah1*$^{+/+}$, n = 4. *Ddah1*$^{-/-}$, n = 4. **p < 0.01. Data are represented as mean values ± SD. **f** Immunofluorescence staining and quantification of OCN positive cells in femora. Scale bars, 100 μm. *Ddah1*$^{+/+}$, n = 5. *Ddah1*$^{-/-}$, n = 5. **p < 0.01. Yellow arrows for OCN positive cells. Data are represented as mean values ± SD. **g** TRAP staining of tibiae from *Ddah1*$^{+/+}$ and *Ddah1*$^{-/-}$ mice. Scale bar, 100 μm. **h** Quantification of osteoclast number of the bone parameter (No. Oc/B.Pm) and osteoclast surface of bone surface (Oc.S/BS). *Ddah1*$^{+/+}$, n = 4. *Ddah1*$^{-/-}$, n = 4. *p < 0.05. Data are represented as mean values ± SD. **i** Representative micro-CT images of trabecular bone and cortical bone in the distal femur (left) from *Ddah2*$^{+/+}$ and *Ddah2*$^{-/-}$ male mice at 12 weeks of age. Scale bars = 200 μm. **j** Bone volume/total volume (BV/TV) and cortical bone thickness (Ct.Th) of *Ddah2*$^{+/+}$ and *Ddah2*$^{-/-}$ mice. *Ddah2*$^{+/+}$, n = 3. *Ddah2*$^{-/-}$, n = 3. Data are represented as mean values ± SD. The data were analyzed by an unpaired two-tailed Student's *t* test in all panels.

used a Flexcell tension system to apply a tension force to the osteoblasts. We found that 5% tension rate force applied to pre-OBs induced DDAH1 expression and reduced the ADMA concentration in the cell supernatant (Supplementary Fig. 4a, b). We next investigated whether the potential mechanical force-response factors including YAP, TAZ, Piezo1, and β-catenin were involved in regulating DDAH1[28–31]. We found that TAZ knockdown most dramatically reduced the Ddah1 expression in stimulation with tension force (Supplementary Fig. 4c, d). Actually, the application of tension force activated YAP/TAZ pathway and increased the expression of ALP (Fig. 5f). Furthermore, the YAP/TAZ antagonist verteporfin (VP) suppressed DDAH1 expression in BMSCs and osteoblasts (Supplementary Fig. 5a). Conversely, the YAP/TAZ agonist LPA induced DDAH1 expression in BMSCs and osteoblasts (Supplementary Fig. 5b).

Using a bioinformatics method, we found that there are no TAZ-binding sites in the promoter region of *Ddah1*[32]. However, the previous study suggested that TAZ cooperates with SMAD4 to regulate osteogenesis[33], and we found that the *Ddah1* promotor region contains several potential SMAD4-binding sites. To confirm the potential underlying mechanism, we performed co-immunoprecipitation (Co-IP) assay and found that tension force promoted the interaction between TAZ and SMAD4 (Fig. 5g). Furthermore, immunofluorescence and nuclear-plasma separation assay demonstrated that tension force promoted the interaction between TAZ and SMAD4 and the nuclear translocation status of these two factors (Fig. 5h and Supplementary Fig. 6). In addition, chromatin immunoprecipitation (ChIP) assay demonstrated that tension force promoted SMAD4 binding to the promoter regions of *Ddah1* (Fig. 5i, j). Accordingly, luciferase reporter assays provided evidence that silencing *Smad4* or *Taz* reduced *Ddah1* transcriptional activity in stimulation with tension force (Fig. 5k). Consistent with the results of tension force treatment, fluid shear stress (FSS) and hard matrix stiffness promoted osteogenesis by analyzing ALP staining and mRNA levels of osteoblast-specific genes (Supplementary Fig. 7a, b, d). Meanwhile, these two different forms of mechanical force also induced DDAH1 expression by activating TAZ/SMAD4 pathway (Supplementary Fig. 7c, e, f). Taken together, these findings indicate that mechanical force activates the TAZ/SMAD4 pathway to induce DDAH1 transcriptional expression.

***Ddah1* deficiency in osteoblast-lineage cells lacked the response to mechanical force.** To further elucidate whether DDAH1 mediated the mechanical force-responsive changes of ADMA in bone, we next subjected *Ddah1*$^{f/f}$ and *Ddah1*$^{Prx1}$ mice to run on the treadmill. As shown in Figs. 4a, b, exercise increased the bone volume/total volume (BV/TV) of trabecular bone in the femurs of

*Ddah1*$^{f/f}$ mice, but the favorable effects were partially impaired by deletion of *Ddah1*, like in *Ddah1*$^{Prx1}$ mice. Meanwhile, the cortical bone thickness (Ct. Th) was neither increased after exercise in *Ddah1*$^{Prx1}$ mice (Fig. 6a, b). Furthermore, exercise increased the number of OCN$^+$ osteoblasts in the femurs of *Ddah1*$^{f/f}$ mice, but not in the *Ddah1*$^{Prx1}$ mice (Fig. 6c, d). In addition, *Ddah1*$^{Prx1}$ mice showed partially impaired MARs after being treated by exercise (Fig. 6e, f). However, ADMA was still in a higher concentration in *Ddah1*$^{Prx1}$ mice after being treated by exercise (Fig. 6g). Collectively, these data suggest that deletion of *Ddah1* in osteoblast-lineage cells leads to a partial loss of response to mechanical force and accordingly regulates bone mass by hydrolyzing ADMA.

**Inhibition of DDAH1 by PD404182 had no effects on preventing bone formation in *Ddah1*$^{-/-}$ mice, but administration of AAV-DDAH1 accelerated bone healing in vivo.** We next sought to understand whether inhibition of DDAH1 could prevent the bone formation in mice. We, therefore, treated the mice with PD404182 (PD), an inhibitor of DDAH1, which is reported in a previous study[34] (Fig. 7a). We confirmed that the administration of PD promoted bone loss in *Ddah1*$^{+/+}$ mice, but the effects of inhibiting bone formation were not found in *Ddah1*$^{-/-}$ mice treated with PD (Fig. 7b). As shown in Fig. 7c–f, micro-CT analysis revealed that the values of BMD, BV/TV, Tb.N and Tb.Th of wild-type mice treatment with PD were markedly lower than *Ddah1*$^{+/+}$ mice treated with vehicle. Furthermore, administration of PD decreased the OCN positive cells of the bone parameter in tibiae of mice (Fig. 7g–k). Conversely, administration of PD had no effects on promoting the loss of OCN positive cells in hindlimbs of *Ddah1*$^{-/-}$ mice. Consistent with the data of immunofluorescence assay, the calcein labeling assay demonstrated that PD remarkably inhibited bone formation in hindlimbs of *Ddah1*$^{+/+}$ mice, but the values of MAR and BFR/BS of PD-treatment *Ddah1*$^{-/-}$ mice were not significantly different from those of vehicle-treatment *Ddah1*$^{-/-}$ mice (Fig. 7h–j). As expected, the concentration of ADMA in serum samples in *Ddah1*$^{-/-}$ mice was notably higher than that of *Ddah1*$^{+/+}$ mice, whereas PD dramatically increased the concentration of ADMA in *Ddah1*$^{+/+}$ mice but had no effects on the concentration of ADMA in *Ddah1*$^{-/-}$ mice (Fig. 7l). In addition, we administrated AAV-DDAH1 to treat bone defects of *Ddah1* conditional knock out mice. The data showed that administration of AAV-DDAH1 dramatically accelerated bone healing in vivo (Fig. 7m, n).

Therefore, these results intimate that inhibition of DDAH1 by administrating of PD 404182 promotes bone loss in wild-type mice, but it does not work in *Ddah1* knock out mice. While administration of AAV-DDAH1 accelerates bone healing in *Ddah1* conditional knock out mice.

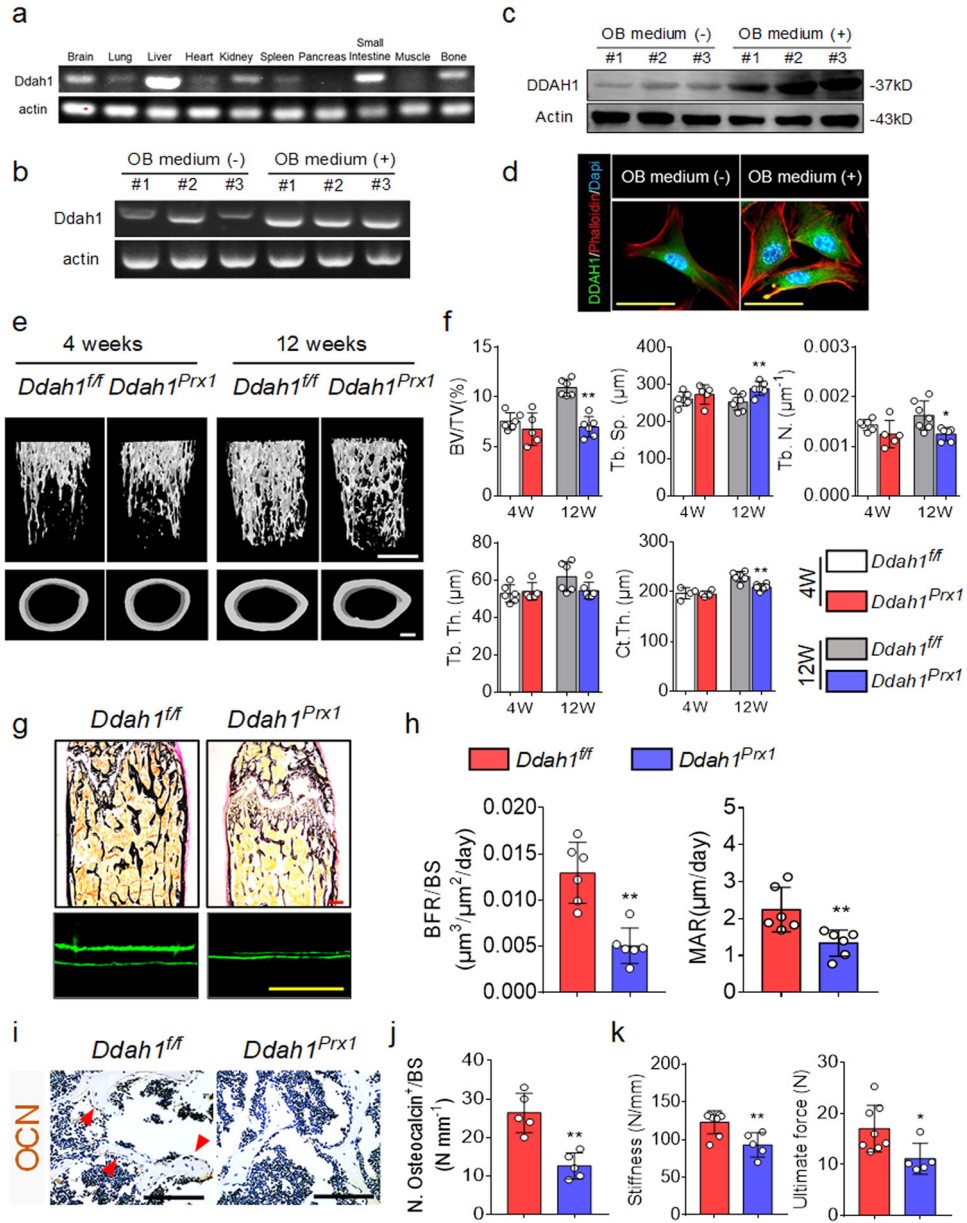

**Fig. 3 Ddah1 deficiency in osteoblast-lineage cells leads to lower bone mass. a** Relative mRNA expression of Ddah1 in several different tissues of C57 mice. **b** RT-PCR assay of Ddah1 in osteoblasts under osteogenic medium. **c** Indicated protein levels of DDAH1 in osteoblasts under osteogenic medium. **d** Immunofluorescence staining of DDAH1 expression in osteoblasts. Scale bar = 100 μm. Green, DDAH1. Red, Phalloidin. Blue, DAPI. **e** Representative micro-CT images of trabecular bone and cortical bone in the distal femur (left) in littermate controls ($Ddah1^{f/f}$, $n = 6$) and osteoblast-specific knock out mice ($Ddah1^{prx1}$, $n = 5$) male mice at 4 weeks of age, and $Ddah1^{f/f}$($n = 6$) compared with $Ddah1^{prx1}$($n = 6$) at 12 weeks of age. **f** Bone volume/total volume (BV/TV), trabecular bone thickness (Tb.Th), trabecular bone number (Tb.N), trabecular bone separation (Tb.Sp), and cortical bone thickness (Ct.Th) of $Ddah1^{f/f}$ and $Ddah1^{prx1}$. 4-week-old $Ddah1^{f/f}$, $n = 6$, $Ddah1^{prx1}$, $n = 5$. 12-week-old $Ddah1^{f/f}$, $n = 6$, $Ddah1^{prx1}$, $n = 6$. *$p < 0.05$. **$p < 0.01$. Data are represented as mean values ± SD. **g** Representative histomorphometric images of the femur and calcein double staining. Scale bars, 100 μm. **h** Quantification of histomorphometric parameters of femur in $Ddah1^{f/f}$ and $Ddah1^{prx1}$ littermates at 12 weeks of age. $Ddah1^{f/f}$, $n = 6$. $Ddah1^{prx1}$, $n = 6$. **$p < 0.01$. Data are represented as mean values ± SD. **i** Representative immunohistochemistry staining images of trabecular bone in $Ddah1^{f/f}$ and $Ddah1^{prx1}$ mice. Red arrows for OCN positive cells. Scale bar = 100 μm. **j** Quantification analysis of OCN⁺ cells in $Ddah1^{f/f}$ and $Ddah1^{prx1}$ mice. $Ddah1^{f/f}$, $n = 5$. $Ddah1^{prx1}$, $n = 5$. **$p < 0.01$. Data are represented as mean values ± SD. **k** Biomechanical analysis of the femurs of $Ddah1^{f/f}$ and $Ddah1^{prx1}$ mice. $Ddah1^{f/f}$, $n = 8$. $Ddah1^{prx1}$, $n = 5$. *$p < 0.05$. **$p < 0.01$. Data are represented as mean values ± SD. The data were analyzed by an unpaired two-tailed Student's $t$ test in two groups compared. One-way analysis of variance (ANOVA) with post-hoc Tukey's test was used for experiments with three or more groups.

## Discussion

In this study, we found that the -394 4N del/ins polymorphism of *Ddah1* was closely associated with low BMD, as well as an increased concentration of ADMA in human serum samples. Deletion of *Ddah1* but not *Ddah2* led to bone loss in mice mainly via impairing bone formation. Meanwhile, an increase of ADMA

regulated by Ddah1 deficiency was contributed to bone loss. In particular, high ADMA accumulation was observed and did not respond to exercise-induced bone formation in Ddah1 conditional knockout mice. Underlying mechanism study unveiled that mechanical force enhanced TAZ/SMAD4 mediated Ddah1 transcription, which in turn regulated ADMA level during bone

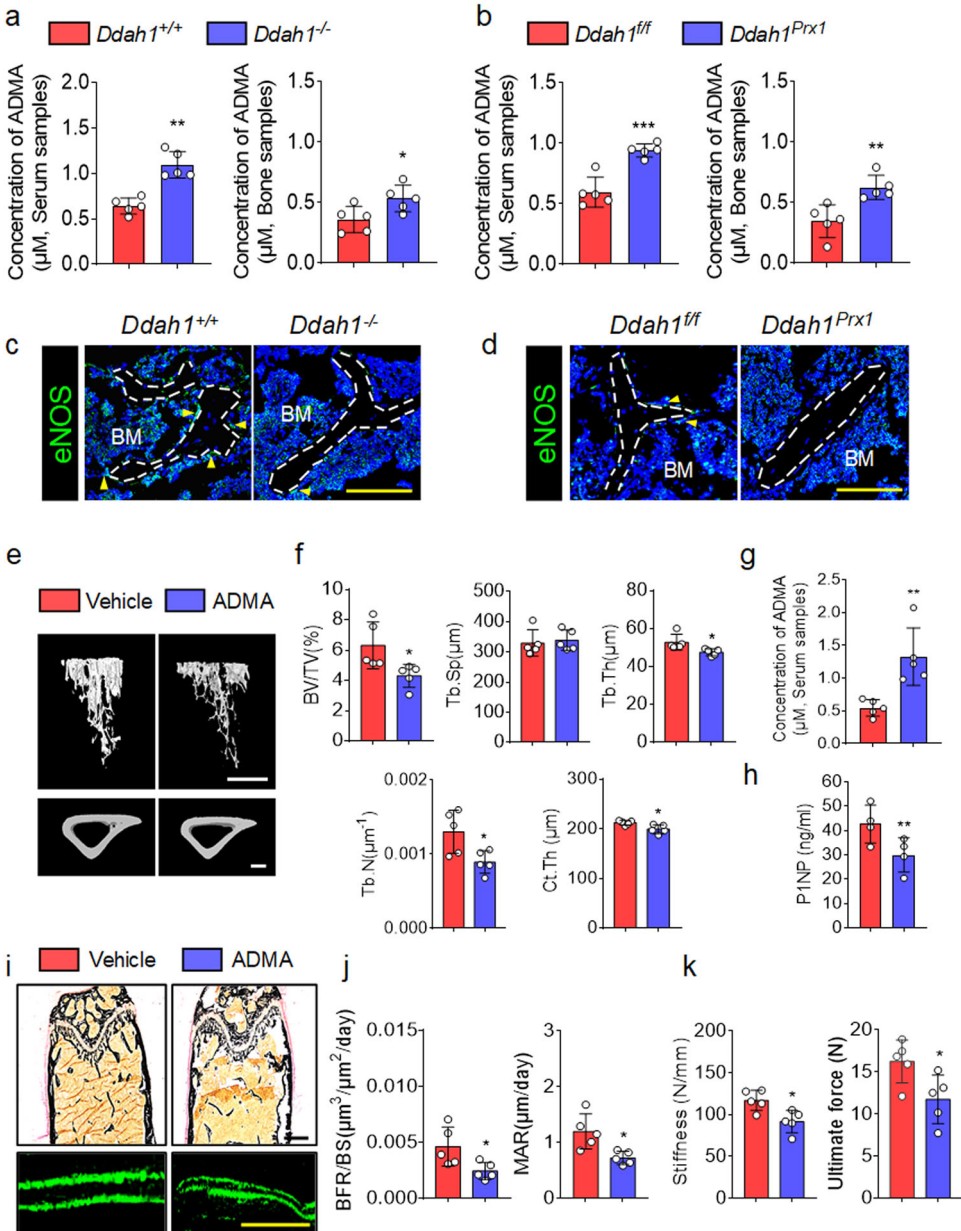

**Fig. 4 Ddah1 deficiency increases concentrations of ADMA to induce bone loss in vivo. a** Serum and bone samples of 10-week old *Ddah1+/+* and *Ddah1−/−* mice were analyzed by LC-MS assay. *Ddah1+/+*, n = 5. *Ddah1−/−*, n = 5. **p < 0.01. ***p < 0.005. Data are represented as mean values ± SD. **b** Serum and bone samples of 12-week-old *Ddah1f/f* and *Ddah1prx1* mice were analyzed by an ADMA specific ELISA assay. *Ddah1f/f*, n = 5. *Ddah1prx1*, n = 5. *p < 0.05. **p < 0.01. Data are represented as mean values ± SD. **c** Immunofluorescence staining of eNOS (green) in bones of *Ddah1+/+* and *Ddah1−/−* mice. Scale bar = 100 μm. **d** Immunofluorescence staining of NOS enzymes in bones of *Ddah1f/f* and *Ddah1prx1* mice. Scale bar = 100 μm. **e** Mice were treated with saline solution or ADMA (12.5 mg/ml), and typical micro-CT images from tibiae are shown. Vehicle-treated group, n = 5, ADMA-treated group, n = 5. **f** Bone-microstructure analysis by micro-CT. The bone volume of total volume (BV/TV), trabecular bone separation (Tb.Sp), trabecular bone thickness (Tb.Th), trabecular bone number (Tb.N), and cortical bone thickness (Ct.Th) values are shown. *p < 0.05. Data are represented as mean values ± SD. **g** The serum ADMA concentrations were confirmed in ELISA assays. Vehicle-treated group, n = 5, ADMA-treated group, n = 5. **p < 0.01. Data are represented as mean values ± SD. **h** Serum P1NP concentrations were confirmed by performing ELISA assays. Vehicle-treated group, n = 4, ADMA-treated group, n = 4. **p < 0.01. Data are represented as mean values ± SD. **i** The BFR/BS ratio and MAR were analyzed by von Kossa staining and double-calcein staining. Scale bar = 100 μm. **j** Quantitative analysis of BFR/BS and MAR values. Vehicle-treated group, n = 5, ADMA-treated group, n = 5. *p < 0.05. Data are represented as mean values ± SD. **k** Bone biomechanics were analyzed by performing a three-point bending mechanics test. The bone stiffness and ultimate force are shown. Vehicle-treated group, n = 5, ADMA-treated group, n = 5. *p < 0.05. Data are represented as mean values ± SD. The data were analyzed by an unpaired two-tailed Student's *t* test in two groups compared.

formation. Finally, inhibition of DDAH1 by PD404182 reduced bone formation in wild-type mice, but administration of AAV-DDAH1 accelerated bone healing in bone defect mice.

Recent discoveries have highlighted the SNPs of *Ddah1* was associated with cardiovascular disease and diabetes[10–12]. In particular, a novel loss-of-function *Ddah1* promoter polymorphism was associated with increased susceptibility to cardiovascular diseases and thrombosis stroke[12]. Thus, these SNPs of *Ddah1* were likely associated with metabolic syndrome (MS), which was regarded as an increased risk factor of low bone

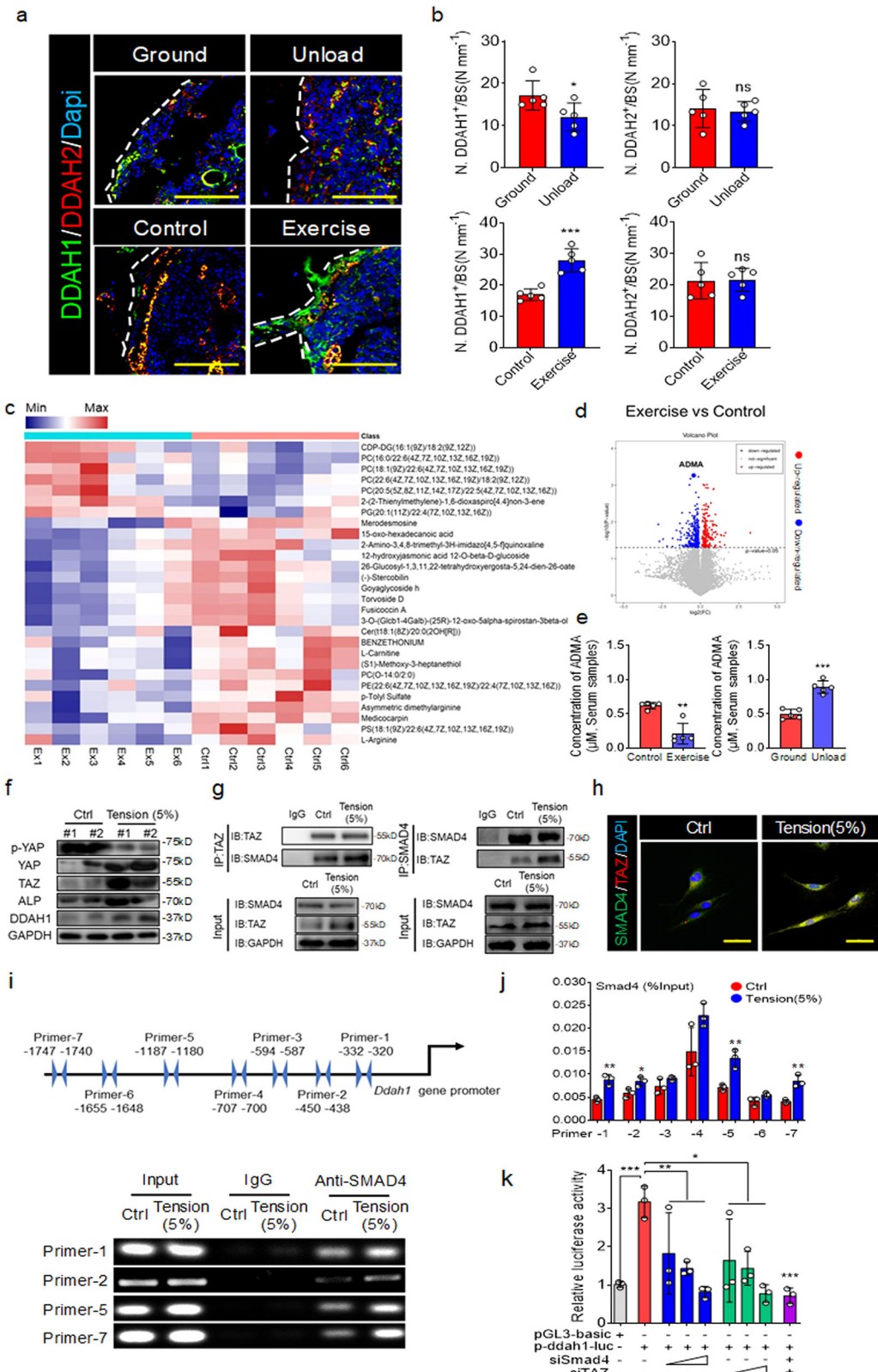

mineral density[14]. However, whether the loss-of-function of *Ddah1* promoter polymorphism being contributed to bone mineral density was not clear. As we have shown, the -394 4N del/ins polymorphism was strongly associated with bone mineral density in a large Chinese population. Meanwhile, the plasma concentration of ADMA was correlated with the -394 4N del/ins polymorphism of *Ddah1*, which was contributed to the

expression of Ddah1. Therefore, these data suggested that loss-of-function DDAH1 promoter polymorphism was a risk factor for bone loss in humans.

To further explore whether DDAH expression was contributed to bone metabolism or not, we firstly generated *Ddah1*$^{-/-}$ mice and *Ddah2*$^{-/-}$ mice. The results suggested that global knockout of *Ddah1* but not *Ddah2* resulted in bone loss in mice mainly via

**Fig. 5 The expression of DDAH1 is responsive to mechanical force via TAZ/SMAD4 signaling pathway. a** The typical images of immunofluorescence staining of DDAH1 (green) and DDAH2 (red) in bones of exercise-treated or tail-suspension mice. Scale bar = 100 μm. **b** Quantification analysis of DDAH1 and DDAH2 expression of bone parameters. ns no significance. Ground group mice, $n = 5$. Unload group mice, $n = 5$. Control group, $n = 5$. Exercise group, $n = 5$. *$p < 0.05$. ***$p < 0.005$. Data are represented as mean values ± SD. **c** Metabonomics of serum samples of control mice and exercise mice. **d** Volcano plot of different metabolites responsive to exercise treatment. **e** Serum samples of exercise mice or tail-suspension mice were analyzed by LC-MS assay. Ground group mice, $n = 5$. Unload group mice, $n = 5$. Control group, $n = 5$. Exercise group, $n = 5$. **$p < 0.01$. ***$p < 0.005$. Data are represented as mean values ± SD. **f** Western blotting analysis of the levels of phosphor-YAP, YAP, TAZ, ALP, and DDAH1 in osteoblasts treated with tension force for 6 h or not, and then cultured for 18 h, the cycle was repeated three times. **g** Co-IP assay of the association between SMAD4 and TAZ in simulation with tension force. **h** Representative immunofluorescence staining images of the association between SMAD4 (green) and TAZ (red) in simulation with tension force. Scale bars = 100 μm. **i** ChIP assay of SMAD4 clustering on the promoter regions of Ddah1 in stimulation with tension force. **j** Quantification analysis of ChIP assay. Control, $n = 3$. Tension, $n = 3$. *$p < 0.05$. **$p < 0.01$. Data are represented as mean values ± SD. **k** Luciferase reporter genes assay of Ddah1 from WT primary osteoblasts in stimulation with tension force by the silence of TAZ and SMAD4. All groups, $n = 3$. *$p < 0.05$. **$p < 0.01$. ***$p < 0.005$. Data are represented as mean values ± SD. The data shown in all panels were analyzed by an unpaired two-tailed Student's $t$ test. One-way analysis of variance (ANOVA) with post-hoc Tukey's test was used for experiments with three or more groups.

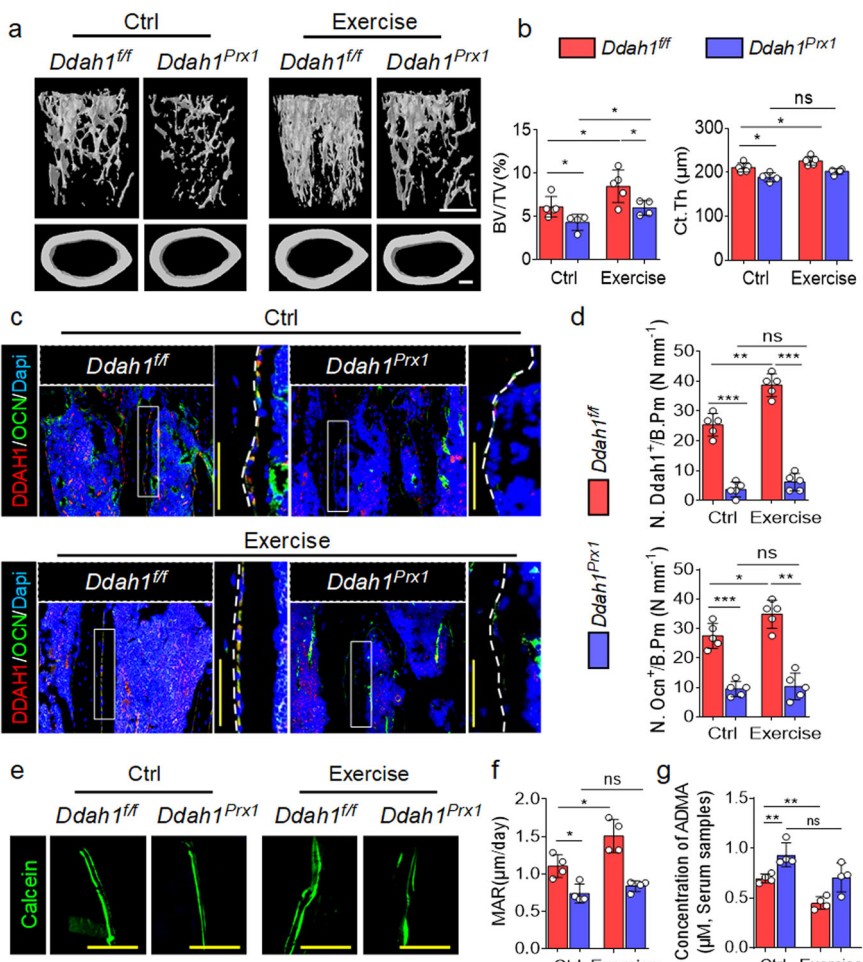

**Fig. 6 *Ddah1* deficiency in osteoblast-lineage cells lacks of the response to mechanical stimulation. a** Representative micro-CT images of trabecular bone and cortical bone in the distal femur (left) of *Ddah1f/f* and *Ddah1prx1* male mice under exercise condition. **b** Bone volume/total volume (BV/TV) and cortical bone thickness (Ct. Th) of *Ddah1f/f* and *Ddah1prx1* mice. *$p < 0.05$. ns, no significance. *Ddah1f/f* control, $n = 5$. *Ddah1f/f* exercise, $n = 5$. *Ddah1prx1* control, $n = 4$. *Ddah1prx1* exercise, $n = 4$. Data are represented as mean values ± SD. **c** Representative immunofluorescence staining images of trabecular bone in *Ddah1f/f* and *Ddah1prx1* mice treated with running or not. Red for DDAH1. Green for OCN. Scale bar=100 μm. **d** Quantification analysis of DDAH1 positive or OCN positive cells in bone parameters. ns, no significance. ns, no significance. *Ddah1f/f* control, $n = 5$. *Ddah1f/f* exercise, $n = 5$. *Ddah1prx1* control, $n = 5$. *Ddah1prx1* exercise, $n = 5$. *$p < 0.05$. **$p < 0.01$. ***$p < 0.005$. Data are represented as mean values ± SD. **e** Representative double-calcein staining images of trabecular bone in *Ddah1f/f* and *Ddah1prx1* mice treated with running or not. Scale bars = 100 μm. **f** MAR of double-calcein staining analysis. ns, no significance. *Ddah1f/f* control, $n = 4$. *Ddah1f/f* exercise, $n = 4$. *Ddah1prx1* control, $n = 4$. *Ddah1prx1* exercise, $n = 4$. *$p < 0.05$. Data are represented as mean values ± SD. **g** Results from LC-MS assay for ADMA concentrations of *Ddah1prx1* and *Ddah1f/f* treated with running or not. ns, no significance. *Ddah1f/f* control, $n = 4$. *Ddah1f/f* exercise, $n = 4$. *Ddah1prx1* control, $n = 4$. *Ddah1prx1* exercise, $n = 4$. *$p < 0.05$. **$p < 0.01$. Data are represented as mean values ± SD. The data were analyzed by an unpaired two-tailed Student's $t$ test in two groups compared. One-way analysis of variance (ANOVA) with post-hoc Tukey's test was used for experiments with three or more groups.

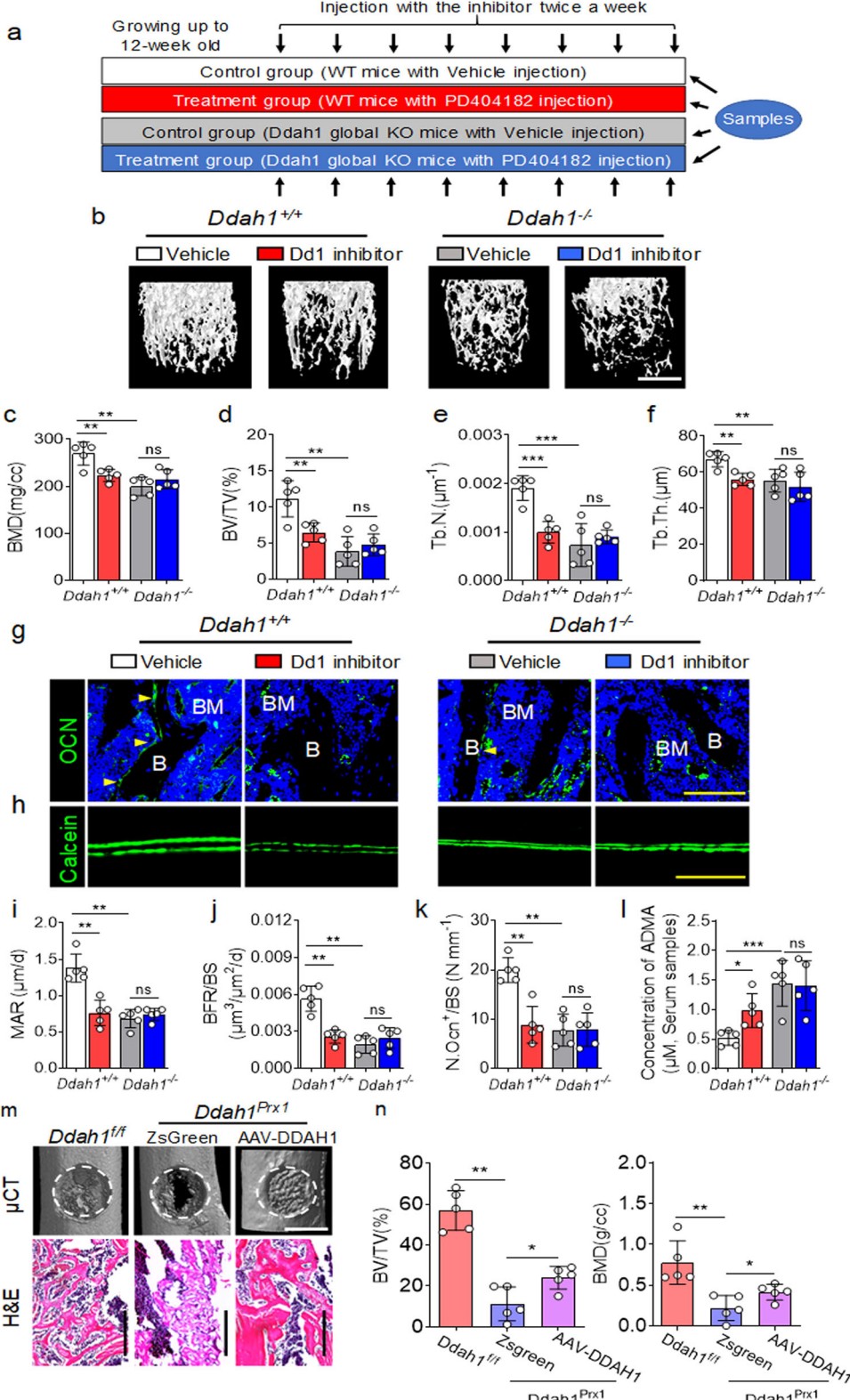

suppressing bone formation, which might be due to the unchanged concentrations of ADMA. We compared the DDAH1 expression in osteoblasts with that in osteoclasts. However, we confirmed that DDAH1 expression was relatively abundant in osteoblasts. Thus, we speculated that the function of osteoclasts was relatively insignificant compared with the function of osteoblasts in *Ddah1*[−/−] mice. Previous studies have focused on

the effects of DDAH1 on brain, kidney and heart tissues[35−37] because DDAH1 has beneficial effects on nitric oxide synthase (NOS) to regulate NO production by degrading ADMA[6]. NO production was also important for bone homeostasis. NOS expression has been examined in osteoblast lineage cells both in vitro and in vivo, and eNOS is the most expressive isoform in bone tissue[38]. It is suggested that deletion of eNOS gene leads to

**Fig. 7 Inhibition of DDAH1 by PD404182 reduces bone formation but administration of AAV-DDAH1 accelerates bone healing in vivo. a** Schematic illustration of in vivo treatment model. **b** Representative micro-CT images of trabecular bone in the distal femur (left) of ground and unload treatment mice administrated by vehicle or PD404182. **c–f** Bone mineral density (BMD), Bone volume/total volume (BV/TV), trabecular bone thickness (Tb.Th) and trabecular bone number (Tb.N) of *Ddah1*[+/+] and *Ddah1*[−/−] mice in the administration of vehicle or PD404182. *Ddah1*[+/+] vehicle-treated, $n = 5$. *Ddah1*[+/+] PD404182-treated, $n = 5$. *Ddah1*[−/−] vehicle-treated, $n = 5$. *Ddah1*[−/−] PD404182-treated, $n = 5$. $*p < 0.05$. $**p < 0.01$. $***p < 0.005$. Data are represented as mean values ± SD. **g** Representative immunofluorescence staining images of trabecular bone in *Ddah1*[+/+] and *Ddah1*[−/−] mice in the administration of vehicle or PD404182. Scale bar = 100 μm. Yellow arrows for OCN positive cells. **h** Representative calcein labeling images of trabecular bone in *Ddah1*[+/+] and *Ddah1*[−/−] mice in the administration of vehicle or PD404182. Scale bar = 100 μm. **i, j** Quantification of the BFR/BS ratio and MAR. *Ddah1*[+/+] vehicle-treated, $n = 5$. *Ddah1*[+/+] PD404182-treated, $n = 5$. *Ddah1*[−/−] vehicle-treated, $n = 5$. *Ddah1*[−/−] PD404182-treated, $n = 5$. $*p < 0.05$. $**p < 0.01$. Data are represented as mean values ± SD. **k** Quantification of the OCN positive cells in bone parameters. *Ddah1*[+/+] vehicle-treated, $n = 5$. *Ddah1*[+/+] PD404182-treated, $n = 5$. *Ddah1*[−/−] vehicle-treated, $n = 5$. *Ddah1*[−/−] PD404182-treated, $n = 5$. $*p < 0.05$. $***p < 0.005$. Data are represented as mean values ± SD. **l** Alteration of serum ADMA levels in *Ddah1*[+/+] and *Ddah1*[−/−] mice in the administration of vehicle or PD404182 determined by LC-MS assay. *Ddah1*[+/+] vehicle-treated, $n = 5$. *Ddah1*[+/+] PD404182-treated, $n = 5$. *Ddah1*[−/−] vehicle-treated, $n = 5$. *Ddah1*[−/−] PD404182-treated, $n = 5$. $**p < 0.01$. $***p < 0.005$. Data are represented as mean values ± SD. **m** Representative micro-CT images of bone defect from femur (left) of *Ddah1*[f/f] and *Ddah1*[prx1] mice administrated by AAV-Zsgreen or AAV-DDAH1. **n** Bone mineral density (BMD) and bone volume/total volume (BV/TV) of bone healing in different groups of mice. *Ddah1*[f/f] AAV-Zsgreen-treated, $n = 5$. *Ddah1*[prx1] AAV-Zsgreen-treated, $n = 5$. *Ddah1*[prx1] AAV-DDAH1-treated, $n = 5$. $*p < 0.05$. $**p < 0.01$. Data are represented as mean values ± SD. The data were analyzed by one-way analysis of variance (ANOVA) with post-hoc Tukey's test.

impaired osteogenesis and bone formation, which implied that loss of NO production leads to a reduction of osteogenesis. As expected, we found that concentrations of ADMA in serum samples of *Ddah1*[−/−] mice and *Ddah1*[Prx1] mice were increased, while concentrations of ADMA in bone samples of *Ddah1*[−/−] mice and *Ddah1*[Prx1] mice were also increased. The results provided evidence that deletion of Ddah1 in osteoblast-lineage cells led to the accumulation of ADMA both in bone marrow niches and peripheral blood. Given the elevation of ADMA concentration contributed to reducing NO production and NO is generally beneficial for osteogenesis, we propose that this downstream mechanism is partially involved in the effects of ADMA on bone. Subsequently, the data suggested that NO production was indeed decreased both in the bones of *Ddah1*[−/−] mice and *Ddah1*[Prx1] mice. Consistent with the previous data, mice bred with a high dose of ADMA reduced bone mass, due to weakened bone formation.

As the previous studies reported, mechanical force regulated NO synthesis to enhance the function of endothelial cells and osteoblasts, and the efficiency of NO production was important for MS and bone diseases[17–19]. While the exercise was confirmed to enhance bone mass and bone strength[39]. However, DDAH1-mediated ADMA hydrolyzation response to mechanical force was still unclear. To address the hypothesis, we treated mice by tail-suspension and running on a treadmill. Intriguingly, we found that only DDAH1 was response to different mechanical treatments with increased levels of ADMA in unloading mice or decreased levels of ADMA in exercise mice. To date, several proteins as key regulators that are response to mechanical forces, such as YAP/TAZ, Piezo1/2, and β-catenin. Laminar shear stress promoted β-catenin nuclear localization in MSCs, which upregulated downstream genes and enhanced osteogenesis[40]. YAP and TAZ are key molecules of the hippo signaling pathway, which are also regarded as mechanosensitive transcription regulators[41]. It has been suggested that the localization and transcriptional activity of YAP and TAZ are regulated by the extracellular matrix (ECM) stiffness and external mechanical force[42]. Other researchers demonstrated that YAP/TAZ signaling was mediated by LATS, which depended on the actin cytoskeleton[43]. Although YAP and TAZ are generally considered as cofactors to regulate cell fate, data from several studies have suggested that they also can function independently. Pan et al.[44] revealed that YAP itself promoted osteogenesis and suppressed adipogenesis by regulating β-catenin signaling. While Byun et al.[45] provided evidence suggesting that Wnt signaling stabilizes TAZ but not YAP to regulate osteogenesis. In addition, YAP nuclear localization is influenced

by Piezo1 in osteoblasts. Piezo1 has been reported to coordinate osteoblast-osteoclast crosstalk through directly responding to mechanical forces in osteoblasts[46]. Meanwhile, Zhou et al.[47] also suggested that Piezo1/2-mediated mechanotransduction was essential for bone formation. In this study, we found that neither YAP nor Piezo1 affects the expression of DDAH1 during stimulation with tension force. However, TAZ played a role in regulating DDAH1 transcription. Previously findings showed that the TAZ/SMAD4 axis played a reciprocal role in promoting osteogenesis, and direct binding to SMAD4 promotes the nuclear retention of TAZ[33]. Interestingly, our data suggest that TAZ recruits SMAD4 to promote nuclear localization of SMAD4 to enhance the transcriptional expression of *Ddah1*. Meanwhile, mechanical forces enhanced the binding capacity between TAZ and SMAD4, as well as transcription of *Ddah1*.

To further confirm whether exercise treatment enhanced bone mass and quality by regulating DDAH1/ADMA pathway, we forced *Ddah1*[f/f] and *Ddah1*[Prx1] mice to run on the treadmill. Exercise dramatically enhanced trabecular bone volume of *Ddah1*[f/f] mice, but slightly enhanced that of *Ddah1*[Prx1] mice. Notably, the cortical thickness of *Ddah1*[f/f] mice was increased by treatment with exercise, but it was not significantly changed in *Ddah1*[Prx1] mice. Consistent with the micro-CT analysis, OCN positive cells and MAR were also not significantly increased in *Ddah1*[Prx1] mice after treatment with running on a treadmill. Meanwhile, the serum level of ADMA was not eliminated by bone tissues in *Ddah1*[Prx1] mice. Thus, our data suggest that deletion of *Ddah1* in osteoblast-lineage cells impairs the ability to hydrolyze ADMA in vivo. On the other hand, tail-suspension treatment-induced bone loss and accumulation of ADMA in vivo. However, inhibition of DDAH1 by intraperitoneal injection with PD404182 promoted bone loss in vivo by inhibiting bone formation, but the effects were not found in *Ddah1*[−/−] mice. The data provided evidence that improving inhibition of DDAH1 could inhibit bone formation in vivo. In addition, administration with AAV-DDAH1 accelerated bone defect healing in *Ddah1*[Prx1] mice compared with AAV-Zsgreen treated mice, which was consistent with the previous data.

To date, the reduction of bone mass due to osteoporosis have been treated by different approaches including bisphosphonates, PTH, denosumab, and romosozumab[48–50] that still have the limitations of treating osteoporosis. Indeed, astronauts and prolonged bed-rest patients are typically bearing serious bone loss and decrease in bone strength[51,52]. Although the underlying mechanism may be complicated, our study at least demonstrated that ADMA and DDAH1 levels are associated with disuse-related

osteoporosis, while deletion of *Ddah1*, a mechanical response gene, is contributed to the reduction of bone formation. Our findings also inform that further studies can be focused on potential therapeutic approaches related to reducing ADMA or enhancing DDAH1 for preventing osteoporosis.

## Methods

**Mice and in vivo treatment**. Generation of $Ddah1^{-/-}$ mice were described previously, a gift from Professor Yingjie Chen, Minnesota University[27]. Mice with *Ddah1* conditionally knocked out (*Ddah1* cKO) in osteoblast lineage cells were generated by crossing male $Ddah1^{f/f}$ mice (a gift from Professor Yingjie Chen, Minnesota University) with female Prx1-cre mice (Jackson Lab #005584). Male $Ddah1^{f/f}$ and $Ddah1^{Prx1}$ mice were sacrificed at 12 weeks of age. Male and female $Ddah2^{+/-}$ mice (C57BL/6N-Ddah2tm1cyagen) were offered by Cyagen Biology Technology. Female wild-type C57BL/6 (B6) mice were from the SLAC Laboratory Animal Company (Shanghai, China). All animal procedures were conducted in compliance with all applicable ethical regulations using procedures approved by the Sir Run Run Shaw Hospital Committee for Animal Resources. The mice were housed with conditions of 12 h dark/12 h light cycle, 22 °C ambient temperature, and 50% humidity. All mice were routinely genotyped using standard PCR protocols.

The compounds used in this study are as follows: NG,NG-dimethyl-L-Arginine (ADMA, APExBIO, C5216), PD 404182 (Ddah1 inhibitor, R&D Systems, #5124). Dosages and time courses are noted in the corresponding text and figure legends.

**Mouse exercise protocol**. Exercise capacity was determined using a treadmill (Life Science, Woodland Hills, CA, USA) running tests as detailed below and previously described[53]. Briefly, exercise group mice were subjected to running on the treadmill at speed of 20 cm/s, 30 min/day for 21 consecutive days. All animal studies were performed according to approved guidelines for the use and care of live animals (Guideline on Administration of Laboratory Animals released in 1988, and 2006 Guideline on Humane Treatment of Laboratory Animals from China). All of the experimental procedures were approved by the Committees of Animal Ethics and Experimental Safety of Sir Run Run Shaw Hospital and Zhejiang Chinese Medical University.

**Tail-suspension mouse model**. Tail-suspension is attempted to achieve an unloading status of hindlimbs. Briefly, the 12-week-old WT mice were individually caged or suspended by the tail with a strip of adhesive surgical tape that was attached to a chain hanging from a pulley. The mice were suspended at a 30° angle to the floor with only the forelimbs touching the floor, which allowed the mice to move and to access food and water freely. The mice were subjected to hindlimb unloading through tail suspension for 28 days. After euthanasia, the bone tissues were collected. All animal studies were performed according to approved guidelines for the use and care of live animals (Guideline on Administration of Laboratory Animals released in 1988, and 2006 Guideline on Humane Treatment of Laboratory Animals from China). All of the experimental procedures were approved by the Committees of Animal Ethics and Experimental Safety of Sir Run Run Shaw Hospital.

**Adeno-associated virus administration model**. When assessing the therapeutic effects of AAV-DDAH1 or AAV-Zsgreen, the gelatin sponge was manually soaked with AAV-DDAH1 or AAV-Zsgreen for 1 h on ice and immediately placed to the bone defection area of femurs in $Ddah1^{f/f}$ and $Ddah1^{Prx1}$ mice. All mice were euthanized by $CO_2$ at time points indicated, the femurs were analyzed by micro-CT and H&E staining.

**Micro-CT and biomechanical testing**. Preparation of bone tissue and micro-CT analysis were performed simply[54]. Bone microarchitecture analysis was performed with μ-QCT system SkyScan1176 (Bruker, Kartuizersweg, Belgium). For analyzing the bone mass of the femur, a region of trabecular bone 2.0 mm wide was contoured, starting 600 microns from the proximal end of the distal femoral growth plate. For femoral trabecular bone, a threshold of 80–255 permille was used. The region of interest of the femoral cortical bone was 1.0 mm wide, starting 3.7 mm from the proximal end of the distal femoral growth plate. The cortical bone was determined in a threshold of 125–255 permille. Three-dimensional reconstructions were created by stacking the two-dimensional images from the indicated regions.

Three-point bending of the femur was conducted to assess bone strength. The femurs were stored in saline at −80 °C. Mechanical testing was performed using a MTS 858 Mini Bionix Biomaterial Testing Machine (MTS, USA). A support span of 5 mm at the bottom of the femur was used, and the load was applied at the midpoint of the posterior aspect of the femur. All tests were performed using a 500-N load cell at a constant loading rate of 6 mm/min.

**Bone histomorphometry**. Mice were injected with 20 mg/kg calcein (Sigma) 10 days and 7 days before euthanisation. Dynamic bone histomorphometry was performed as described[55]. In brief, the femurs were fixed in 4% PBS-buffered paraformaldehyde and dehydrated in an ascending ethanol series. Subsequently, bones were embedded in methacrylate and cut into 7-μm sections to assess the fluorescent calcein labels. Unstained sections were analysed using fluorescence microscopy to determine the mineral apposition rate (MAR), and the bone formation rate/bone surface (BFR/BS), as well as the bone volume/total volume (BV/TV), trabecular number (Tb.N), trabecular separation (Tb.Sp), and trabecular thickness (Tb.Th). To determine the number of osteoclasts, the femur was decalcified for 2 weeks using 10% EDTA (Merck, Shanghai, China), dehydrated, and embedded into paraffin. Bone sections were analysed using the Osteomeasure software (Osteometrics, USA) following international standards. For Vonkossa staining, mice limbs were not decalcified, they were embedded into methacrylate and cut into 5-μm-thick sections.

**Immunofluorescence assay**. Immunofluorescence was mainly performed as described[56]. Briefly, freshly dissected bones were fixed in 4% paraformaldehyde for 48 h and incubated in 15% DEPC-EDTA (pH 7.8) for decalcification. Then, specimens were embedded in paraffin or OCT and sectioned at 8 μm. Sections were blocked in PBS with 10% horse serum for 1 h and then stained overnight with mouse-anti-Osteocalcin (Santa Cruz, 1:100, sc-376726), rabbit-anti-Ddah1 (SAB, 1:200, #37368), mouse-anti-Ddah1 (Santa Cruz, 1:100, sc-271337), rabbit-anti-Ddah2 (SAB, 1:200, #38934), mouse-anti-TAZ (Abcam, 1:200, ab242313), rabbit-anti-YAP (Abcam, 1:200, ab52771), and eNOS (Santa Curz, 1:200, sc-376751). Goat-anti-mouse FITC (1:1000; Jackson ImmunoResearch, 705-165-147) and donkey-anti-rabbit Alexa Fluor 488 (1:1000; Molecular Probes, A21206) were used as secondary antibodies. DAPI (Cell Signaling Technology, #4083) and DyLight™ 594 Phalloidin (Cell Signaling Technology, #12877) were used for counterstaining. All immunofluorescence experiments were confirmed by at least one independent repeat. An Olympus IX81 confocal microscope or Zeiss LSM-880 confocal microscope was used to image samples.

**Mice serum samples analysis**. The bone turnover marker pro-collagen type I N-terminal peptide (P1NP) were measured in the serum using ELISA kits (Elabscience Biotechnology, Wuhan, China) according to the manufacturer's instructions. Serum ADMA and bone tissue ADMA concentrations were determined by LC-MS assay.

**Cell culture and in vitro treatment**. Primary murine pre-osteoblasts were isolated from calvarial cells of fetal mice, while bone marrow mesenchymal stem cells (BMSCs) were extracted from femurs and tibiae of 6-week-old mice. The marrow plug was flushed by using syringe with a range of 1 mL. These plugs were then dispelled into single cell and were seeded in a 10-cm dish containing α-MEM (Corning, New York, USA) supplemented with 10% fetal bovine serum (FBS).

For osteogenesis, mature osteoblasts were differentiated using the standard osteogenic medium in α-MEM with 10% FBS, 50 μg/ml L-ascorbic acid, and 1080 mg/ml β-glycerophosphate, 1% penicillin/streptomycin. The osteoblast differentiation test was performed as the osteoblastgenesis protocol, as the previous study described[57]. For quantitative analysis of Alp activity, osteoblasts were incubated with Alamar Blue and were then incubated with phosphatase substrate (Sigma-Aldrich, St. Louis, MO) dissolved in 6.5 mM Na2CO3, 18.5 mM NaHCO3, 2 mM MgCl2. Alp activity was then read with a spectrophotometer (Thermo Scientific, Shanghai, China). Bone nodule formation was stained with 1 mg/mL Alizarin Red S solution (pH 5.5) after 21 days of induction. RNA was isolated at various time points as indicated.

For mechanical stimulation, the elastic cell culture plates were firstly rinsed by PBS, and then covered by 100 mg/ml poly-lysine solution for 2 h. Next, the cells were seeded into the plates at a 90% confluence. After adhered, we started the instruments (Mechanical Cell Strain Instrument, Tianjin Technology University; Flexcell FX-5000T Tension SYSTEM, USA) to set a 5% of tension rate with a cycling loading stimulation. The drugs and compounds used in this study are as follows: as lysophosphatidic acid (LPA, Sigma-Aldrich, L7260), verteporfin (VP, Selleck, S1786), NG,NG-dimethyl-L-Arginine (ADMA, APExBIO, C5216). Dosages and time courses are noted in the corresponding text and figure legends.

**siRNA-mediated gene silencing, and Cre retrovirus transfection**. Negative control siRNA and oligo-targeting siRNAs were transfected into pre-osteoblasts using Lipofectamine RNAiMAX (Life Technologies, Shanghai, China) at 50 nM of siRNA in 24-well plates, 12-well plates or cell culture compartments following the manufacturer's instructions. In brief, the cells were incubated for overnight with 1.5 mL of Opti-MEM (GIBCO) containing Lipofectamine RNAiMAX per $0.18 \times 10^6$ cells and 50 nM siRNA. Experiments and mRNA analyses were conducted 48 h later.

GFP control retrovirus and Cre retrovirus (HANBIO, Shanghai, China) were infected into pre-osteoblasts in six-well plates or cell culture compartments following the manufacturer's instructions. The cells were infected for 48 h with retrovirus. Subsequent experiments were performed after 48 h.

**RNA isolation, reverse transcription, and real-time PCR**. Total RNA was extracted using TRIzol reagent (Invitrogen) or RNeasy Mini Kit (Qiagen), and reverse transcription was performed with the High-Capacity cDNA Reverse

Transcription Kit from Applied Biosystems according to the manufacturer's instructions. We performed quantitative analysis of gene expression using SYBR® Green PCR Master 878 Mix (Applied Biosystems) with the LightCycler 480 real-time PCR system (Roche Life Science, China). Gapdh expression was used as an internal control. The sequence of the primers used for PCR is listed in Supplementary table 3.

**Western blot assay**. Western blot assay was performed according to previously described standard protocol. Primary antibodies were specific for DDAH1 (1:1000; SAB, #37368), phospho-YAP (1:1000; Cell Signaling Technology, #4911), YAP (1:1000; Cell Signaling Technology, #14074), TAZ (1:1000; Cell Signaling Technology, #83669), ALP (1:500; Santa Cruz, #sc-271431), SMAD4 (1:1000; Cell Signaling Technology, #46535), GAPDH (1:5000; Proteintech, #60004-1-Ig), alpha-tubulin (1:5000; Proteintech, #66031-1-Ig), and Histone H3 (1:5000; Proteintech, #17168-1-AP). Secondary anti-mouse/rabbit HRP-conjugated antibodies were subsequently applied.

**Co-immunoprecipitation assay**. As performed in our previous study, briefly, cell extracts were first precleared with 25 μL of protein A/G-agarose (50% v/v). The supernatants were immunoprecipitated with 2 μg of anti-TAZ antibodies for overnight at 4 °C, followed by incubation with protein A/G-agarose for 4 h at 4 °C. Protein A/G-agarose-antigen-antibody complexes were collected by centrifugation at 300 g for 60 s at 4 °C. The pellets were washed five times with 1 mL IPH buffer (50 mM Tris-HCl, pH 8.0, 150 mM NaCl, 5 mM EDTA, 0.5% Nonidet P-40, 0.1 mM PMSF), for 5 min each time at 4 °C. Bound proteins were resolved by SDS-PAGE, followed by western blotting with the anti-TAZ, anti-SMAD4 or anti-GAPDH antibodies. The experiments were replicated at least three times.

**Chromatin immunoprecipitation assay**. Transcription factor binding sites within −2000 bps before the murine Ddah1 coding start site was searched using TFSEARCH software to identify putative Ddah1 binding sites. ChIP assays were performed to test for binding of Smad4 to each of the seven Ddah1 binding sites, following our published procedure[58]. Briefly, the sheared chromatin from control and tensile-treatment pre-OBs that had been fixed with 1% formaldehyde was immunoprecipitated with antibodies to SMAD4, or rabbit IgG as a negative control. The precipitated DNA was used as a template for PCR using primers specifically designed to amplify a segment of 100–200 bps containing the putative Ddah1 binding sites. The sequences of the primers are listed in Supplementary table 4.

**Luciferase reporter gene assay**. Luciferase reporter gene assay was performed as the instruction indicated (Beyotime Biotechnology #RG028, Shanghai, China). Briefly, pre-OBs were plated in 24-well plates in triplicate, and the cells were transfected with different concentrations of pGL3-basic or pGL3-Ddah1-Luc. After 24 h, the transfected cells were lysed with the lysis buffer (Promega, Madison, WI, USA). Renilla luciferase expression was determined, and luciferase activity was measured using a luciferase assay system (Promega, USA).

**Metabolomics sample preparation, quality control, data extraction and analysis**
*Sample preparation*. Samples were prepared using the automated MicroLab STAR system from Hamilton Company. The sample extracts were stored overnight under nitrogen before preparation for further experiments.

*QA/QC*. Several types of control were analyzed in concert with the experimental samples: a pooled matrix sample generated by taking a small volume of each experimental sample served as a technical replicate throughout the data set, extracted water samples served as process blanks, and a cocktail of QC standards that were carefully chosen not to interfere with the measurement of endogenous compounds was spiked into every analyzed sample, allowed instrument performance monitoring and aided chromatographic alignment. Overall process variability was determined by calculating the median RSD for all endogenous metabolites (non-instrument standards) present in 100% of the pooled matrix samples. Experimental samples were randomized across the platform run with QC samples spaced evenly among the injections.

*Ultrahigh performance liquid chromatography-tandem mass spectroscopy (UPLC-MS/MS)*. All methods utilized a Waters ACQUITY ultra-performance liquid chromatography and a Thermo Scientific Q-Exactive high resolution/accurate mass spectrometer interfaced with a heated electrospray ionization (HESI-II) source and Orbitrap mass analyzer operated at 35,000 mass resolution. The sample extract was dried then reconstituted in solvents compatible to each of the four methods. Each reconstitution solvent contained a series of standards at fixed concentrations to ensure injection and chromatographic consistency. The extract was gradient eluted from a C18 column (Waters UPLC BEH C18-2.1 × 100 mm, 2.5 μm) using water and methanol, containing 0.05% perfluoropentanoic acid (PFPA) and 0.1% formic acid (FA). Another aliquot was also analyzed using acidic positive ion conditions, it was optimized for more hydrophobic compounds. In this method, the extract was

gradient eluted from the same aforementioned C18 column using methanol, acetonitrile, water, 0.05% PFPA and 0.01% FA and was operated at an overall higher organic content. Another aliquot was analyzed using basic negative ion optimized conditions using a separate dedicated C18 column. The basic extracts were gradient eluted from the column using methanol and water, with 6.5 mM Ammonium Bicarbonate at pH 8. The fourth aliquot was analyzed via negative ionization following elution from a HILIC column (Waters UPLC BEH Amide 2.1 × 100 mm, 2.5 μm) using a gradient consisting of water and acetonitrile with 10 mM Ammonium Formate, pH 10.8. The MS analysis alternated between MS and data-dependent MSn scans using dynamic exclusion. The scan range varied slightly between methods but covered 70–1000 m/z.

*Data preprocessing and statistical analysis*. The acquired LC-MS raw data were analyzed by the progenesis QI software (Waters Corporation, Milford, USA) using the following parameters. The precursor tolerance was set at 5 ppm, fragment tolerance was set at 10 ppm, and retention time (RT) tolerance was set at 0.02 min. Internal standard detection parameters were deselected for peak RT alignment, isotopic peaks were excluded for analysis, and noise elimination level was set at 10.00, the minimum intensity was set to 15% of base peak intensity. The Excel file was obtained with 3D data sets including m/z, peak RT and peak intensities, and RT-m/z pairs were used as the identifier for each ion. The resulting matrix was further reduced by removing any peaks with a missing value (ion intensity = 0) in more than 50% of samples. The internal standard was used for data QC (reproducibility). Metabolites were identified by progenesis QI (WatersCorporation, Milford, USA) Data Processing Software, based on public databases and self-built databases. The positive and negative data were combined to get a combined data which was imported into R ropls package. Principle component analysis (PCA) and (orthogonal) partial least-squares-discriminant analysis (O)PLS-DA were carried out to visualize the metabolic alterations among experimental groups, after mean centering (Ctr) and Pareto variance (Par) scaling, respectively. The Hotelling's T2 region, shown as an ellipse in score plots of the models, defines the 95% confidence interval of the modeled variation. Variable importance in the projection (VIP) ranks the overall contribution of each variable to the OPLS-DA model, and those variables with VIP > 1 are considered relevant for group discrimination. In this study, the default 7-round cross-validation was applied with 1/seventh of the samples being excluded from the mathematical model in each round, in order to guard against overfitting. The differential metabolites were selected on the basis of the combination of a statistically significant threshold of variable influence on projection (VIP) values obtained from the OPLS-DA model and p values from a two-tailed Student's t test on the normalized peak areas, where metabolites with VIP values >1.0 and p values <0.05 were considered as differential metabolites.

All reagents were analytical or HPLC grade, which were from CNW Technologies GmbH (Düsseldorf, Germany). L-2-chlorophenylalanine was from Shanghai Hengchuang Bio-technology Co., Ltd. (Shanghai, China).

**Collection and determination of human serum and hemocytes sample**. Venous blood samples were taken from the cubital vein of patients. The sampling was performed between 7:00 a.m. and 3:00 p.m. with the majority of samples being taken in the mornings before 12:00 a.m. Blood samples were centrifuged for dividing serum aliquots and hemocytes. Both of them were stored at −80 °C. The information of patients was provided in Supplementary table 2. In brief, all patients and controls were of Han Chinese ancestry, and the three different groups were determined by T values (T ≥ −1.0 to control group, −1.0 > T > −2.5 to osteopenia group, and T ≤ −2.5 to osteoporosis group).

Serum samples were determined on the LC-MS assay to investigate the concentrations of ADMA, ADMA standard was used as a control, and we determined used SDMA standard to distinguish the differences between ADMA and SDMA. In our analysis, the linear associated results of ADMA and SDMA are good in the range of 0.1–1000 ng/ml. The QC results of serum samples and standard samples were verified.

DNA samples were extracted from hemocytes, and the characterizes of individuals were provided in Supplementary table 1. Then, the polymorphism genotyping analysis was performed to determine the −394 4N del/ins polymorphism of Ddah1 (Probe sequences were listed in Supplementary table 5) by LightCycler 480 real-time PCR system (Roche Life Science, China), and SNaPshot assay (Supported by GENESKY, Shanghai, China).

**Statistical and reproducibility**. Each experiment was performed at least three times, data are presented as mean ± standard deviation (S.D.). The variance was similar between groups for most parameters assessed. The normality of data was determined using the Kolmogorov-Smirnov test. In cases where data were normally distributed, statistical evaluations of two-group comparisons were performed using a two-sided Student's t test. One-way analysis of variance (ANOVA) was used for experiments with three or more groups. Two-way ANOVA with Bonferroni post hoc tests was used for analyzing genotype and treatment effects. If data were not normally distributed, the Mann–Whitney test and the Wilcoxon signed-rank test were used for data analysis. Logistic regression was used to test for genetic association in human samples with and without adjusting for sex, age and body mass index. Graphs and statistics were prepared using GraphPad Prism 8.0 software.

Differences were considered significant at $P < 0.05$ (*$P < 0.05$, **$P < 0.01$, ***$P < 0.001$).

**Study approval**. All animal studies were performed according to approved guidelines for the use and care of live animals (Guideline on Administration of Laboratory Animals released in 1988, and 2006 Guideline on Humane Treatment of Laboratory Animals from China). All of the experimental procedures were approved by the Committees of Animal Ethics and Experimental Safety of Sir Run Run Shaw Hospital, Zhejiang University. The human study was approved by the Medical Ethics Committees of Sir Run Run Shaw Hospital, and written informed consents were obtained from the participants before venous blood samples collection.

**Reporting summary**. Further information on research design is available in the Nature Research Reporting Summary linked to this article.

## Data availability
The authors declare that the data supporting the findings of this study are available within the article and its Supplementary information or Source data file. Metabolomics was provided in MetaboLights, accession code is MTBLS3725 (https://www.ebi.ac.uk/metabolights/MTBLS3725/descriptors). Source data are provided with this paper as Supplementary Information file. Source data are provided with this paper.

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

## Acknowledgements

We thank Pro. Yingjie Chen (Minnesota University, MN55455, USA) for providing *Ddah1*[+/−] and *Ddah1*[f/f] mice. We thank the animal core facility of Sir Run Run Shaw Hospital and GemPharmatech Biopharmaceutical for assistance. We also thank Shanghai Applied Protein Technology and Shanghai Luming Biological Technology for assistance of metabonomics analysis. Kindly thank Pro. Xin Dong from Fudan University for assistance of LC-MS assay. This work was supported in part by grants from, the National Natural Science Foundation of China (81972089, 81871797, 81972504, and 81802680), National Key R&D Program of China (2018YFC1105202), The Key cultivation Project of National Natural Science Foundation of China (92068102), and The Key Research and Development Plan in Zhejiang Province (2018C03060), Key Project of Natural Science Foundation of Zhejiang Province (Z20H060003).

## Author contributions

Conceptualization: S.F., A.Q., and Z.X. Data curation: Z.X., S.S., and A.Q. Funding acquisition: S. F., S.S., and X.F. Investigation: Z.X., L.H., S.S., Y.W., J.W., Z.J., X.L., X.Z., Q.M., S.W., Y.H., J.C., and P.Y. Methodology: X.Z., X.Z., W.X., H.W., and J.C. Project administration: Z.X., S.F., and A.Q. Resources: L.H., A.Q., L.N., Z.X., W.X., and P.Y. Supervision: S.F., X.F., and A.Q. Visualization: X.Z. and H.W. Writing-original draft: Z.X. and A.Q. Writing-review & editing: A.Q., S.F., and X.F.

## Competing interests

The authors declare no competing interests.
