## [Peer Review File · Nature Communications]

Reviewer comments

Reviewer #1 (Remarks to the Author):

This manuscript by Xie et al. aimed to study the role of Ddah1 in bone formation. The authors showed that Ddah1 deficiency leads to the decrease of bone formation. The study of Ddah1 function in bone is well designed and the in vivo results are convincing. In addition, they investigated the role of Ddah1 in exercise mediated bone formation and observed that Ddah1 is transcriptionally stimulated by Smad4-TAZ in response to mechanical stress in vitro. However, more evidence is needed to support the mechanism of exercise mediated Ddah1 stimulation

1. It is unclear that tension induces the interaction of Smad4 and TAZ and nuclear localization of them. In Fig 5g, the result of immunoprecipitation with anti-Smad4 antibody will further support their conclusion. In Fig 5h, biochemical study such as cytosol and nuclear fractionation in control and tension induced cell lysates will provide another evidences. In Fig 5i, Chromatin IP with Anti-TAZ antibody will be another evidence of authors conclusion.

2. Authors used Flexcell tension system for exercise mimetic in vitro system. They tried to knockdown YAP, TAZ, Piezo1 and β -catenin with siRNA. In supplement Fig 2c, they observed that TAZ siRNA alone significantly reduces Ddah1 expression. However, it would be essential to verify the knockdown of YAP, TAZ, Piezo1 and β -catenin in the experimental condition. In addition, authors observed that YAP is not involved in tension mediated Ddah1 expression, though it is known that YAP also interacts with Smad4 for transcriptional regulation. Thus, to understand a different functional role of TAZ and YAP in tension signal mediated Ddah1 expression, it would be essential to provide a further evidence that YAP is not involved in Ddah1 expression through Chromatin IP and reporter gene analysis.

3. In supplement Fig 3a, quantified ALP activity should be added in both Ad-GFP and Ad-Cre condition. In supplement Fig 3b, results of gene expression of Ade-Cre infected Ddah1f/f osteoblasts are missing. These results should be compared with the results of gene expression of Ade-GFP infected Ddah1f/f osteoblasts. Also, in supplement Fig 3c, authors need to add TAZ and smad4 level in static and FSS condition of Ade-Cre infected Ddah1f/f osteoblasts. In supplement Fig 3e, to further understand mechanical signal to Ddah1, it is important whether TAZ and Smad4 nuclear localization is altered in 2 and 40 kPa of Ade-Cre infected Ddah1f/f osteoblasts. These results will provide another evidences that Ddah1, as a target of TAZ-Smad4, is critical for osteoblast differentiation.

(minor comments)

1. Authors need to check labels in supplement Fig 3e. FSS or ##kPa???
2. line 326 "It is suggested that deletion of eNOS gene leads to impaired osteogenesis and bone formation, which implied that loss of NO production leads to a reduction of osteogenesis [39]." Authors need to check whether cited reference is correct.

Reviewer #2 (Remarks to the Author):

General Comments

The goal of this manuscript was to evaluate the role of ADMA and Ddah1 in bone formation and adaptation. In addition, the authors investigated the mechanism to how this ADMA/Ddah1 interaction affects bone formation. The results suggested that mechanical force activates the TAZ/SMAD4 pathway to induce Ddah1 expression, which decreases ADMA synthesis to enhance bone formation. While the effect of mechanical loading inducing bone formation has been known for some time now, exploring the role of ADMA and Ddah1 signaling in bone formation is new. The authors conducted an extensive analysis of these signaling pathways including knockout mice models, clinical gene analysis, and in-vitro mechanism studies. The presentations of the data are structured very well. However, several concerns were raised in the version of the paper.

1. The authors could highlight the novelty behind their study more in the introduction. i.e. how does looking into ADMA/Ddah1 pathway advance the field of mechanobiology and what are the implications in the clinic?
2. The methodology for the enrollment of subjects is unclear. The authors did not state what the inclusion/exclusion criteria was or how they determined which individuals had osteopenia and osteoporosis.
3. The authors did not provide any evidence of the knockout mouse models used in this study were successful in knocking out the genes indicated. Validation results should be included in a supplementary figure.
4. The authors noted that osteoclast surface/bone surface fraction increased following knockout of Ddah1, but did not further look into this aspect besides the expression of Ddah1 in osteoclasts, which the data was not shown. An in-vitro study analyzing the effect of osteoclast activity from Ddah1 knockout could enhance the mechanism proposed in this study.
5. The methodology for the bone defect mouse model was not stated in the paper, which needs to be included. In addition, the use of AAV-DDAH1 was not introduced in the paper, which raises the question as to why it was used over MS023 to accelerate bone healing.

Specific Comments (Line=Ln, Page = Pg)

Abstract

Pg. 2, Ln. 44: Suggested to change "response" to "respond."

Introduction

Pg. 3, Ln. 67: Suggested to change Ddah1 to DDAH1 to be consistent with previous statements. Need to introduce the SNP abbreviation before.

Pg. 3, Ln. 76: Suggested to change "response" to "responsive."

Results

Pg. 7, Ln. 179: What bone tissues were you analyzing? Could be more specific.

Pg. 7, Ln. 194: There is no data representing a knockout of eNOS. Was this supposed to state "knockout of Ddah1"?

Pg. 8, Ln. 207: Suggested to change "shown" to "shows".

Pg. 8, Lns. 209-212: Suggested to condense statement to highlight results.

Pg. 8, Lns. 215-217: Are there any significant results worth reporting about Piezo1 and β -catenin?

Discussion

Pg. 11, Lns. 314-317: Consider to reword sentence for clarity.

Pg. 11, Lns. 317-319: Was there significant DDAH1 expression in osteoclasts? Unclear based on this statement. May want to consider adding a supplementary figure to present this data.

Pg. 12, Ln. 326: Suggested to change "expression" to "expressed."

Pg. 12, Lns. 343-345: Suggested to remove "whether" and change "responsive" to "response."

Pg. 12, Ln. 346: Suggested to change "response" to "responsive."

Methods

Pg. 24, Lns. 699-702: What was the frequency of the cyclic loading and what was the duration of the loading? Need to be more specific.

Reviewer #3 (Remarks to the Author):

This is a study on the importance of asymmetric dimethylarginine in the regulation of bone mass and the skeletal response to exercise. The large amount of human, mouse and in vitro data are shown. Overall, the results support the hypothesis that asymmetric dimethylarginine plays a role in determining osteoblast function. Some aspects of the study require clarification.

Specific Comments

1. The description of the human studies is inadequate.
 - a. No methodological description of the human studies is provided. It is not clear how participants were classified into the 'control', 'osteopenia' and 'osteoporosis' groups. Presumably DXA was used, but this is not described.
 - b. There is also no mention of ethics approval for the human studies. Did participants provide informed consent?
 - c. The statistical section also does not refer to the human studies. It is not clear how the adjustment was performed on the data in Figure 1d.
2. The manuscript claims numerous times that Ddah1 deficiency leads to bone loss. However, bone loss means that bone mass decreases from a higher value to a lower value. Consequently, bone mass can only be investigated by doing longitudinal bone mass studies. So such studies seem to have been performed. As such, there is no documentation of bone loss in these data.
3. It is claimed that Ddah1 knockout leads to 'dramatically decreased bone formation'. However, this seems to be inconsistent with the very small effect of Ddah1 knockout on distal femur BV/TV (Figure 2b). This inconsistency should be addressed.

Reviewer #4 (Remarks to the Author):

This is the report of a series of experiments and translational analyses in a human cohort that analyzes the involvement of DDAH1 and ADMA in bone formation. The strength of this study lies in its combination of multiple experimental settings in animal models to address the relationship between ADMA metabolism and bone formation, e.g., unloading, exercise, gene deletion and pharmacological enzyme inhibition. Whilst the data reported are potentially important and interesting, there are a number of flaws to the study that the authors need to address. Also, the human cohort that was included in this manuscript is insufficiently characterized, and the presentation of results from the human study is unclear.

Major concerns:

1. The authors have used homozygous Ddah1 knockout mice that have been well characterized before with respect to residual DDAH activity and ADMA concentrations in various tissues and blood. They also used homozygous Ddah2 knockout mice that were crossbred from heterozygous parents that are commercially available. Although they genotyped the mice to confirm the knockout, no

information is given regarding ADMA concentrations in blood plasma and tissues for this mouse line. This is needed to be able to interpret the role of DDAH2 and ADMA in bone formation, as the differential roles of both DDAH isoenzymes in ADMA metabolism is being debated.

2. The authors included a large sample of human subjects in their study. As they report, 1,404 individuals were genotyped for the -394 4N del/ins polymorphism in the DDAH1 gene. Blood samples from 570 of these individuals were used for analyzing ADMA plasma concentration by LC-MS. Purportedly, these individuals were grouped into those with normal BMD, osteopenia, or osteoporosis. The manuscript lacks information on why ADMA was not analyzed in the whole group of patients, how the subgroup was selected, how BMD, osteopenia, and osteoporosis were diagnosed, and how many of the 570 were categorized into the three subgroups according to bone health. Also, the frequency of the genotypes for the DDAH1 polymorphism in each of the subgroups should be included. Ideally, a table should be provided comprising all of this information on the human cohort.

3. The table presented in Figure 1 d cannot be interpreted properly as is. The table lists “frequency” in the subgroups of individuals with normal BMD, osteopenia, and osteoporosis, Frequency of what? Of the homozygous ins/ins, heterozygous del/ins, or homozygous del/del genotype? Then, what do odds ratios signify? As presented, these are odds ratios for the presence of the polymorphism in the subgroups of individuals. If so, this is taken the wrong way: Odds ratios should be presented for the presence of a clinical condition in individuals with different genotypes, i.e. the odds ratios of osteopenia or osteoporosis in carriers of the DDAH polymorphism.

Also, in figure 1a-c, individual data points should be plotted rather than coloured areas, to better visualize the number of potential outliers in each of the subgroups.

4. It is completely unclear why the authors report that “The data showed that deletion of DDAH1 in osteoblasts reduced the level of eNOS in bones, which was related to the accumulated ADMA.” (lines 184-185). Do the authors have data to support their view that deletion of Ddah1, by increasing ADMA, also reduces the protein expression of eNOS? If so, this would be important information, as such a mechanism has never been shown before. Current general understanding is that ADMA inhibits the enzymatic activity of NOS, and this – if at all – would compensatorily cause upregulation of NOS expression rather than downregulation. If the finding of reduced eNOS protein expression in bone is real (as it seems from the immunofluorescence photographs), this might independently add to the effect of Ddah1 deletion on bone structure. The authors should take an effort to dissect the two mechanisms and their respective influence on bone structure, DDAH expression and ADMA on the one hand and eNOS expression on the other hand.

5. In using a PRMT inhibitor like MS023, the authors need to confirm that the observed effects depend on reduced ADMA accumulation rather than on modified protein function secondary to reduced posttranslational methylation. Protein methylation has been shown to affect important cellular processes like RNA transcription, cell cycle regulation, and intracellular signaling. How can the authors exclude that one or several of these mechanisms caused the changes in bone structure that they observed after MS023 treatment?

Further, MS023 has been reported to be a PRMT1, 3, and 6 inhibitor (e.g., Samuel et al., *Proteomes* 2018), not a specific PRMT1 inhibitor; thus, the authors should note that the specific enzyme responsible for ADMA formation in bone remains unclear.

6. ADMA was measured by LC-MS in the human samples but by ELISA in the mouse samples. Taking into consideration that it is much harder to reliably extract metabolites from tissues as compared to plasma, the authors should consider re-analyzing mouse plasma and tissue samples by LC-MS. As such, ELISA has a higher degree of variability and less specificity than LC-MS analysis.

7. In Figure 5c, the authors show that exercise in mice reduced both, the levels of ADMA and those of L-arginine. As a net effect, therefore, substrate availability for NOS should have remained

unchanged, and – thus – NOS activity as well. Can the authors provide evidence for changed NO generation in bone in exercised vs. sedentary mice?

8. To the same end, physical activity should be included as a co-variate in statistical analysis in the human cohorts.

Minor issues:

1. The English language needs some revision.
2. ADMA concentrations should be re-calculated to $\mu\text{mol/L}$ to make them more easily comparable with previous published studies.
3. In line 123 the text should read “-394 4N ...”
4. Lines 171 and 192 should read “mechanical properties”
5. Line 396 should read “complex”
6. For reference 52 there are no authors listed.

REVIEWER COMMENTS

Reviewer #1 (Remarks to the Author):

This manuscript by Xie et al. aimed to study the role of Ddah1 in bone formation. The authors showed that Ddah1 deficiency leads to the decrease of bone formation. The study of Ddah1 function in bone is well designed and the in vivo results are convincing. In addition, they investigated the role of Ddah1 in exercise mediated bone formation and observed that Ddah1 is transcriptionally stimulated by Smad4-TAZ in response to mechanical stress in vitro. However, more evidence is needed to support the mechanism of exercise mediated Ddah1 stimulation

1. It is unclear that tension induces the interaction of Smad4 and TAZ and nuclear localization of them. In Fig 5g, the result of immunoprecipitation with anti-Smad4 antibody will further support their conclusion. In Fig 5h, biochemical study such as cytosol and nuclear fractionation in control and tension induced cell lysates will provide another evidences. In Fig 5i, Chromatin IP with Anti-TAZ antibody will be another evidence of authors conclusion.

Thanks for your comments! As suggested, we performed the immunoprecipitation with anti-Smad4 antibody, and added the results into our Fig 5. Furthermore, we provided the results of cytosol and nuclear fractionation treated by tension in the Supplement figure 4. However, we found that TAZ might be unable to directly bind to the sites of Ddah1 promoter, but as mentioned in the text, Smad4 has various binding sites on the region of Ddah1 promoter.

2. Authors used Flexcell tension system for exercise mimetic in vitro system. They tried to knockdown YAP, TAZ, Piezo1 and β -catenin with siRNA. In supplement Fig 2c, they observed that TAZ siRNA alone significantly reduces Ddah1 expression. However, it would be essential to verify the knockdown of YAP, TAZ, Piezo1 and β -catenin in the experimental condition. In addition, authors observed that YAP is not involved in tension mediated Ddah1 expression, though it is known that YAP also interacts with Smad4 for transcriptional regulation. Thus, to understand a different functional role of

TAZ and YAP in tension signal mediated Ddah1 expression, it would be essential to provide a further evidence that YAP is not involved in Ddah1 expression through Chromatin IP and reporter gene analysis.

Thanks for your comments! Following the suggestion, we provided the results of western blot assay to verify the efficiency of knock-down by siRNA. Meanwhile, according to the bioinformatic analysis, we did not find YAP is involved in the potential binding sites in the region of Ddah1 promoter as the JASPAR shown, but we tried to perform the reporter gene analysis and found that silence of Yap had no effects on the Ddah1 reporter gene expression (Figure R1).

Figure R1.

3. In supplement Fig 3a, quantified ALP activity should be added in both Ad-GFP and Ad-Cre condition. In supplement Fig 3b, results of gene expression of Ade-Cre infected Ddah1f/f osteoblasts are missing. These results should be compared with the results of gene expression of Ade-GFP infected Ddah1f/f osteoblasts. Also, in supplement Fig 3c, authors need to add TAZ and smad4 level in static and FSS condition of Ade-Cre infected Ddah1f/f osteoblasts. In supplement Fig 3e, to further understand mechanical signal to Ddah1, it is important whether TAZ and Smad4 nuclear localization is altered in 2 and 40 kPa of Ade-Cre infected Ddah1f/f osteoblasts. These results will provide another evidences that Ddah1, as a target of TAZ-Smad4, is critical for osteoblast differentiation.

Thanks for your kindly critical comments! We performed the quantification of ALP activity in the study, and the results were added into the Supplement Fig 3a. Furthermore, we added the data of gene expression of Ad-Cre infected osteoblasts in the Supplement Fig 3b, and the results suggested that knock-down the expression of *Ddah1* inhibited mechanical force-induced the expression of osteogenesis genes. Next, the experiments were analyzed by using *Ddah1*^{fl/fl} derived pre-osteoblasts, and we have confirmed that *Ddah1* transcriptional expression was regulated by TAZ-Smad4, also we provided evidences that FSS and hard stiffness treatment increased the expression of DDAH1 by activating TAZ-Smad4 signaling. Thus, we thought that TAZ-Smad4 signaling was not mainly determined by *Ddah1* expression.

(minor comments)

1. Authors need to check labels in supplement Fig 3e. FSS or ##kPa???

Thanks for your suggestion! We revised the labels in Fig S3e.

2. line 326 “It is suggested that deletion of eNOS gene leads to impaired osteogenesis and bone formation, which implied that loss of NO production leads to a reduction of osteogenesis [39].” Authors need to check whether cited reference is correct.

Thanks for your comments! We examined the cited reference, and the new reference was added in the text.

Reviewer #2 (Remarks to the Author):

General Comments

The goal of this manuscript was to evaluate the role of ADMA and *Ddah1* in bone formation and adaptation. In addition, the authors investigated the mechanism to how this ADMA/*Ddah1* interaction affects bone formation. The results suggested that mechanical force activates the TAZ/SMAD4 pathway to induce *Ddah1* expression, which decreases ADMA synthesis to enhance bone formation. While the effect of mechanical loading inducing bone formation has been known for some time now,

exploring the role of ADMA and Ddah1 signaling in bone formation is new. The authors conducted an extensive analysis of these signaling pathways including knockout mice models, clinical gene analysis, and in-vitro mechanism studies. The presentations of the data are structured very well. However, several concerns were raised in the version of the paper.

1. The authors could highlight the novelty behind their study more in the introduction. i.e. how does looking into ADMA/Ddah1 pathway advance the field of mechanobiology and what are the implications in the clinic?

Thanks for your valuable and thoughtful comments! As we known, it has not been reported that how the metabolites response to mechanical force. In our study, we firstly investigated that Ddah1/ADMA pathway response to mechanical force, which regulates bone formation. The findings broadened the knowledge of both mechanobiology and bone biology. Related Ddah1-targeted drugs could be potential therapeutic approaches for treating bone-related diseases.

Following the suggestion, we revised the part of introduction in the text.

2. The methodology for the enrollment of subjects is unclear. The authors did not state what the inclusion/exclusion criteria was or how they determined which individuals had osteopenia and osteoporosis.

Thanks for your comments! Postmenopausal women and over 50-year old men were involved in the study. We divided the individuals into control group, osteopenia group and osteoporosis group by BMD (T values, $T \geq -1.0$, $-1.0 > T > -2.5$, $T \leq -2.5$) according to the WHO criteria.

3. The authors did not provide any evidence of the knockout mouse models used in this study were successful in knocking out the genes indicated. Validation results should be included in a supplementary figure.

Thanks for your comments! We performed Western blot assay to analyze the efficiency of knock-out Ddah1 in osteoblast-lineage cells, the data was shown in Supplementary

figure 1.

4. The authors noted that osteoclast surface/bone surface fraction increased following knockout of *Ddah1*, but did not further look into this aspect besides the expression of *Ddah1* in osteoclasts, which the data was not shown. An in-vitro study analyzing the effect of osteoclast activity from *Ddah1* knockout could enhance the mechanism proposed in this study.

Thanks for your suggestion! We performed the in-vitro study to analyze the osteoclast activity from *Ddah1* knockout mice, and we found that *Ddah1* deficiency promoted osteoclastogenesis with M-CSF and RANKL treatment (Figure R2a). However, we compared the expression of DDAH1 in OCs with in OBs, we unexpectedly found that the expression of DDAH1 in OBs was more abundant than in OCs (Figure R2b). Thus, we thought the role of DDAH1 in OBs might be more important than the role of DDAH1 in OCs. Meanwhile, this study was intended to investigate the mechanism between mechanical force and osteogenesis mediated by DDAH1. So we mainly focus on osteoblast at this stage.

Figure R2.

5. The methodology for the bone defect mouse model was not stated in the paper, which needs to be included. In addition, the use of AAV-DDAH1 was not introduced in the paper, which raises the question as to why it was used over MS023 to accelerate bone

healing.

Thanks for your comments! We added the methodology in the paper. In this part, we used Ddah1 conditional knock out mice, we were intended to investigate the therapeutic effects of DDAH1 overexpression on bone healing. Although MS023 administration had effects on preventing bone loss in unload mice by inhibiting ADMA, we further considered that MS023 is a PRMTs inhibitor, which might affect RNA transcription, cell cycle regulation, et al. Therefore, instead of MS023, we used PD404182 (a Ddah1 inhibitor) to perform an additional animal experiment to investigate the effects of DDAH1 on bone formation, and the results of Figure 7 showed that inhibition of Ddah1 by PD had no effects on promoting the bone loss in Ddah1 knock out mice.

Specific Comments (Line=Ln, Page = Pg)

Abstract

Pg. 2, Ln. 44: Suggested to change “response” to “respond.”

Thanks for the suggestion! We revised it in our text.

Introduction

Pg. 3, Ln. 67: Suggested to change Ddah1 to DDAH1 to be consistent with previous statements. Need to introduce the SNP abbreviation before.

Thanks for your kind consideration! We revised the description in the text.

Pg. 3, Ln. 76: Suggested to change “response” to “responsive.”

Thanks for your suggestion! We changed it in our text.

Results

Pg. 7, Ln. 179: What bone tissues were you analyzing? Could be more specific.

Thanks for your comments! Actually, we analyzed the tibiae by micro-CT. We added the details in our text.

Pg. 7, Ln. 194: There is no data representing a knockout of eNOS. Was this supposed to state “knockout of Ddah1”?

Thanks for your comments! There is no data representing knock out of eNOS, we reworded the statement in our text.

Pg. 8, Ln. 207: Suggested to change “shown” to “shows”.

Thanks for your suggestion! We changed the words.

Pg. 8, Lns. 209-212: Suggested to condense statement to highlight results.

Thanks for your suggestion! We revised the statement in our text.

Pg. 8. Lns. 215-217: Are there any significant results worth reporting about Piezo1 and β -catenin?

Thanks for your comments! The question was also we cared. As the data shown, we found that silencing Piezo1 or β -catenin had no effects of the expression of Ddah1 in the stimulation with mechanical force, at least in the condition of tension force (Supplementary figure 2).

Discussion

Pg. 11, Lns. 314-317: Consider to reword sentence for clarity.

Pg. 11, Lns. 317-319: Was there significant DDAH1 expression in osteoclasts? Unclear based on this statement. May want to consider adding a supplementary figure to present this data.

As the data shown, the expression of DDAH1 was lower in osteoclasts than that in osteoblasts. Thus, we mainly focus on the DDAH1 in osteoblasts, especially in condition of mechanical force (Page 5, Figure 1b).

Pg. 12, Ln. 326: Suggested to change “expression” to “expressed.”

Thanks for your suggestion! We changed the words in the text.

Pg. 12, Lns. 343-345: Suggested to remove “whether” and change “responsive” to “response.”

Thanks for your suggestion! We changed the words in the text.

Pg. 12, Ln. 346: Suggested to change “response” to “responsive.”

Thanks for your suggestion! We changed the words in the text.

Methods

Pg. 24, Lns. 699-702: What was the frequency of the cyclic loading and what was the duration of the loading? Need to be more specific.

Thanks for your comments! We added the details of the cyclic loading on osteoblasts in the text.

Reviewer #3 (Remarks to the Author):

This is a study on the importance of asymmetric dimethylarginine in the regulation of bone mass and the skeletal response to exercise. The large amount of human, mouse and in vitro data are shown. Overall, the results support the hypothesis that asymmetric dimethylarginine plays a role in determining osteoblast function. Some aspects of the study require clarification.

Specific Comments

1. The description of the human studies is inadequate.
 - a. No methodological description of the human studies is provided. It is not clear how participants were classified into the ‘control’, ‘osteopenia’ and ‘osteoporosis’ groups. Presumably DXA was used, but this is not described.

The classification of different groups was determined by T values ($T \geq -1.0$ for control

group, $-1.0 > T > -2.5$ for osteopenia group, $T \leq -2.5$ for osteoporosis group). Postmenopausal women and over 50-year old men without severe diseases were involved in the study. We added the description in our supplementary materials.

b. There is also no mention of ethics approval for the human studies. Did participants provide informed consent?

Thanks for your comments! We have the ethics approval for the human studies in our institute, and we have the informed consent from patients. It was added in the section of study approval.

c. The statistical section also does not refer to the human studies. It is not clear how the adjustment was performed on the data in Figure 1d.

Thanks for your comments! The details of adjustment were presented in Figure 1d (the bottom of table).

2. The manuscript claims numerous times that Ddah1 deficiency leads to bone loss. However, bone loss means that bone mass decreases from a higher value to a lower value. Consequently, bone mass can only be investigated by doing longitudinal bone mass studies. So such studies seem to have been performed. As such, there is no documentation of bone loss in these data.

Thanks for your comments! We have realized that the description was not appropriate. Thus, to show the findings, we reworded as low bone mass instead of bone loss in our text.

3. It is claimed that Ddah1 knockout leads to ‘dramatically decreased bone formation’. However, this seems to be inconsistent with the very small effect of Ddah1 knockout on distal femur BV/TV (Figure 2b). This inconsistency should be addressed.

Thanks for your valuable and thoughtful comments! The sentences were reworded, Ddah1 knockout leads to significantly decreased bone formation, but it seems to not be

a dramatical decrease. Following the suggestion, we addressed the issue.

Reviewer #4 (Remarks to the Author):

This is the report of a series of experiments and translational analyses in a human cohort that analyzes the involvement of DDAH1 and ADMA in bone formation. The strength of this study lies in its combination of multiple experimental settings in animal models to address the relationship between ADMA metabolism and bone formation, e.g., unloading, exercise, gene deletion and pharmacological enzyme inhibition. Whilst the data reported are potentially important and interesting, there are a number of flaws to the study that the authors need to address. Also, the human cohort that was included in this manuscript is insufficiently characterized, and the presentation of results from the human study is unclear.

Major concerns:

1. The authors have used homozygous Ddah1 knockout mice that have been well characterized before with respect to residual DDAH activity and ADMA concentrations in various tissues and blood. They also used homozygous Ddah2 knockout mice that were crossbred from heterozygous parents that are commercially available. Although they genotyped the mice to confirm the knockout, no information is given regarding ADMA concentrations in blood plasma and tissues for this mouse line. This is needed to be able to interpret the role of DDAH2 and ADMA in bone formation, as the differential roles of both DDAH isoenzymes in ADMA metabolism is being debated.

Thanks for your valuable and thoughtful comments! According to the micro-CT assay, Ddah2 knockout did not affect the value of BV/TV compared with wild-type mice. To further investigate it, we analyze the serum samples of homozygous Ddah2 knockout mice by LC-MS, but we found that ADMA concentrations were not upregulated compared with wild-type control mice. So, we hypothesized that the negative phenotype of bone in this Ddah2 knockout mice line might be due to the unchanged concentrations of ADMA.

2. The authors included a large sample of human subjects in their study. As they report,

1,404 individuals were genotyped for the -394 4N del/ins polymorphism in the DDAH1 gene. Blood samples from 570 of these individuals were used for analyzing ADMA plasma concentration by LC-MS. Purportedly, these individuals were grouped into those with normal BMD, osteopenia, or osteoporosis. The manuscript lacks information on why ADMA was not analyzed in the whole group of patients, how the subgroup was selected, how BMD, osteopenia, and osteoporosis were diagnosed, and how many of the 570 were categorized into the three subgroups according to bone health. Also, the frequency of the genotypes for the DDAH1 polymorphism in each of the subgroups should be included. Ideally, a table should be provided comprising all of this information on the human cohort.

Thanks for your comments! In our study, we firstly investigated serum samples of 570 individuals by LC-MS analysis, and we found that ADMA concentrations were upregulated in the osteopenia and osteoporosis groups (groups determined by DXA, $T \geq -1.0$, $-1.0 > T > -2.5$, $T \leq -2.5$). However, the data of -394 4N del/ins polymorphism in Ddah1 gene showed no significant differences in these 570 individuals. To further investigate the relationship, we enlarged the number of individuals to 1404. After further analyzed, we confirmed that the data of -394 4N del/ins polymorphism in Ddah1 gene was related to bone health in the control, osteopenia and osteoporosis groups.

3. The table presented in Figure 1 d cannot be interpreted properly as is. The table lists “frequency” in the subgroups of individuals with normal BMD, osteopenia, and osteoporosis, Frequency of what? Of the homozygous ins/ins, heterozygous del/ins, or homozygous del/del genotype? Then, what do odds ratios signify? As presented, these are odds ratios for the presence of the polymorphism in the subgroups of individuals. If so, this is taken the wrong way: Odds ratios should be presented for the presence of a clinical condition in individuals with different genotypes, i.e. the odds ratios of osteopenia or osteoporosis in carriers of the DDAH polymorphism.

Also, in figure 1a-c, individual data points should be plotted rather than coloured areas, to better visualize the number of potential outliers in each of the subgroups.

Thanks for your comments! We performed a retrospective study that frequency was

represented for the ratio of -394 4N del/ins and ins/ins in control, osteopenia and osteoporosis groups. Odds ratios were presented for the association of -394 4N del/ins or ins/ins with osteopenia and osteoporosis. OR>1 meant that 4N del/ins and ins/ins were risk factors for osteopenia and osteoporosis. Thus, the risk of osteopenia and osteoporosis was positively associated with del/ins and ins/ins. Following the suggestion, we changed figure 1a-c with individual data points by using plotted graphics.

4. It is completely unclear why the authors report that “The data showed that deletion of DDAH1 in osteoblasts reduced the level of eNOS in bones, which was related to the accumulated ADMA.” (lines 184-185). Do the authors have data to support their view that deletion of Ddah1, by increasing ADMA, also reduces the protein expression of eNOS? If so, this would be important information, as such a mechanism has never been shown before. Current general understanding is that ADMA inhibits the enzymatic activity of NOS, and this – if at all – would compensatorily cause upregulation of NOS expression rather than downregulation. If the finding of reduced eNOS protein expression in bone is real (as it seems from the immunofluorescence photographs), this might independently add to the effect of Ddah1 deletion on bone structure. The authors should take an effort to dissect the two mechanisms and their respective influence on bone structure, DDAH expression and ADMA on the one hand and eNOS expression on the other hand.

Thank you for your valuable and thoughtful comments! Firstly, we performed the experiments to confirm that SNAP (donor of NO) reversed the inhibition of osteogenesis induced by Ddah1 knockout in pre-osteoblasts (Figure R3). The results implied that NO pathway was mediated by Ddah1 expression in osteoblasts at least.

To the question of reducing expression of eNOS, we hypothesized that the reducing expression of eNOS in the parameters of bone was due to the decreasing number of osteoblasts. Indeed, knock out of Ddah1 led to the decreasing number of osteoblasts. While, we also investigated the expression of eNOS in vitro, as the Figure R4 shown, knock out of Ddah1 had no effects on the expression of eNOS in osteoblasts.

Figure R3.

Figure R4.

5. In using a PRMT inhibitor like MS023, the authors need to confirm that the observed effects depend on reduced ADMA accumulation rather than on modified protein function secondary to reduced posttranslational methylation. Protein methylation has been shown to affect important cellular processes like RNA transcription, cell cycle regulation, and intracellular signaling. How can the authors exclude that one or several of these mechanisms caused the changes in bone structure that they observed after MS023 treatment?

Further, MS023 has been reported to be a PRMT1, 3, and 6 inhibitor (e.g., Samuel et al., Proteomes 2018), not a specific PRMT1 inhibitor; thus, the authors should note that the specific enzyme responsible for ADMA formation in bone remains unclear.

Thanks for your valuable and thoughtful comments! Actually, we agree with that MS023 is a PRMTs inhibitor, which might affect RNA transcription, cell cycle regulation, et al. However, as the data shown, MS023 significantly downregulated ADMA concentrations at least, reduced ADMA accumulation prevented the bone loss of unloading mice (MS023+ADMA). Further deeper studies need to be performed to certify the role of ADMA in bone structure, instead of the other mechanisms.

To further investigate the role DDAH1/ADMA in bone remodeling, we performed the animal experiments with PD404182 treatment (a DDAH1 inhibitor). We found that PD treatment reduced the bone mass of distal femurs in wild-type mice, but this effect

was not achieved in Ddah1 knockout mice. Meanwhile, bone formation was also inhibited by PD treatment in wild-type mice, but it was not happened in Ddah1 knockout mice.

6. ADMA was measured by LC-MS in the human samples but by ELISA in the mouse samples. Taking into consideration that it is much harder to reliably extract metabolites from tissues as compared to plasma, the authors should consider re-analyzing mouse plasma and tissue samples by LC-MS. As such, ELISA has a higher degree of variability and less specificity than LC-MS analysis.

Thanks for your comments! We performed the LC-MS to re-analyze the mouse plasma and tissue samples. The results were shown in the figures.

7. In Figure 5c, the authors show that exercise in mice reduced both, the levels of ADMA and those of L-arginine. As a net effect, therefore, substrate availability for NOS should have remained unchanged, and – thus – NOS activity as well. Can the authors provide evidence for changed NO generation in bone in exercised vs. sedentary mice?

Thanks for your comments! We agree with that arginine is a source of NO synthesis. To further investigate the effects of exercise in mice, we performed experiments to detect the NO generation in bone of exercised and sedentary mice. As the data shown, we found that exercise promoted the NO generation in bone compared with that of sedentary mice (Figure R5). The results suggested that although both the levels of ADMA and arginine was reduced, the final effect of exercise in mice was to promoting NO generation, which revealed that ADMA mainly.

Figure R5.

8. To the same end, physical activity should be included as a co-variate in statistical analysis in the human cohorts.

Thanks for your valuable and thoughtful comments! Unfortunately, it's difficult to collect the information of physical activity and made a quantification analysis of it in our patients. We did not collect the data of physical activity, and we did not analyze it as a co-variate.

Minor issues:

1. The English language needs some revision.

Thanks for your suggestion! We improved the writing with the help of an English native speaking professor.

2. ADMA concentrations should be re-calculated to $\mu\text{mol/L}$ to make them more easily comparable with previous published studies.

Thanks for your suggestion! The concentrations of ADMA were re-calculated to μM in the experiments.

3. In line 123 the text should read “-394 4N ...”

Thanks for your comments! We revised it.

4. Lines 171 and 192 should read “mechanical properties”

Thanks for your comments! We rewriting the context by “mechanical properties”

5. Line 396 should read “complex”

Thanks for your comments! Following the suggestion, we revised it.

6. For reference 52 there are no authors listed.

Thanks for your comments! We addressed the missing context.

Reviewer comments, further review

Reviewer #3 (Remarks to the Author):

The criticism raised in response to the first version of the manuscript has been addressed.

Reviewer #4 (Remarks to the Author):

The authors have addressed all of the major and minor concern raised in their rebuttal letter to the reviewers. However, some of the major comments have not led the authors to amend the manuscript text or figures; therefore, it is requested here that the manuscript be accordingly amended to match what the authors present in their rebuttal to the reviewers.

Major concerns

1) The differential roles of DDAH1 and DDAH2 were discussed; the authors claim to have performed measurements of ADMA in DDAH2 ko mice, but the data are not presented in the revised manuscript nor in the supplemental data. They need to be included. The Discussion should be amended to include the hypothesis that "the negative phenotype of Ddah2 knockout mice might be due to unchanged concentrations of ADMA", as argued in the rebuttal to the reviewers; page and line where this has been included should be indicated in the response to reviewers.

2) The details of the human study were requested by this reviewer; the authors replied in detail in the rebuttal. No changes are indicated for the manuscript where to find the corresponding amendments in the manuscript. This must be done, and indicated by page and line in the response to reviewers.

3) The figure 1 and table presenting the results of the human study were unclear. The authors' explanations in the rebuttal have shed more light on this to help understand the selection of subgroups, the associations of genotype with bone phenotype etc. However, once again, the footnote to the table does not include an explanation what "Odds ratio" refers to. Also, numbers of patients should be indicated for each group presented in this combined figure and table, to help readers understand more transparently sample sizes.

4) It was criticized by this reviewer that it is implausible to suggest that Ddah1 knockout resulted in decreased eNOS expression. The authors responded to this in detail in their response to reviewers. However, when the reviewer checked the wording of the manuscript (lines 187-189) has remained unchanged and, thus, still completely unclear. There needs to be clarity in the wording to dissect whether the authors suggest that Ddah1 knockout caused elevated ADMA, which then caused impaired NO production, or whether they suggest that Ddah knockout caused reduced eNOS protein expression.

5) The data on PRMT inhibition were discussed by this reviewer previously. The additional experiments using a DDAH inhibitor appear more helpful to dissect the mechanism of the diverse findings presented in this present manuscript. However, the author spent quite some time searching for the MS023 data in the revised manuscript, because the authors did not indicate in their response that this data had been deleted.

7) In this criticism of the reviewer, the lack of data showing the biochemical consequences of reduced ADMA AND reeducated arginine by exercise (Figure 5) were discussed. The authors provided additional measurement results to suggest that NO production was increased by exercise. However, once again, this data is not included in the revised manuscript, so the information is not accessible for the reader. The data must be included in the revised figure or in a supplementary figure.

Reviewer 4:

The authors have addressed all of the major and minor concern raised in their rebuttal letter to the reviewers. However, some of the major comments have not led the authors to amend the manuscript text or figures; therefore, it is requested here that the manuscript be accordingly amended to match what the authors present in their rebuttal to the reviewers.

Major concerns

1) The differential roles of DDAH1 and DDAH2 were discussed; the authors claim to have performed measurements of ADMA in DDAH2 ko mice, but the data are not presented in the revised manuscript nor in the supplemental data. They need to be included. The Discussion should be amended to include the hypothesis that "the negative phenotype of Ddah2 knockout mice might be due to unchanged concentrations of ADMA", as argued in the rebuttal to the reviewers; page and line where this has been included should be indicated in the response to reviewers.

Thanks for your considerable comments! We added the data of ADMA in our supplementary data (supplementary figures 1). Then, the hypothesis was added in our text (page 5, 138-140).

2) The details of the human study were requested by this reviewer; the authors replied in detail in the rebuttal. No changes are indicated for the manuscript where to find the corresponding amendments in the manuscript. This must be done, and indicated by page and line in the response to reviewers.

Thanks for your comments! We made the changes in the supplementary information, please refer to supplementary information page 9.

3)The figure 1 and table presenting the results of the human study were unclear. The authors' explanations in the rebuttal have shed more light on this to help understand the selection of subgroups, the associations of genotype with bone phenotype etc. However, once again, the footnote to the table does not include an explanation what

"Odds ratio" refers to. Also, numbers of patients should be indicated for each group presented in this combined figure and table, to help readers understand more transparently sample sizes.

Thanks for your valuable suggestion! We added the explanation of 'Odds ratio' in the text, it means 'An indicator of the degree of association between exposure and disease'.

Meanwhile, we added the details of number sizes in our text.

4)It was criticized by this reviewer that it is implausible to suggest that Ddah1 knockout resulted in decreased eNOS expression. The authors responded to this in detail in their response to reviewers. However, when the reviewer checked the wording of the manuscript (lines 187-189) has remained unchanged and, thus, still completely unclear. There needs to be clarity in the wording to dissect whether the authors suggest that Ddah1 knockout caused elevated ADMA, which then caused impaired NO production, or whether they suggest that Ddah knockout caused reduced eNOS protein expression.

Thanks for your kind consideration! We further checked the words, and we changed the description in our text.

5)The data on PRMT inhibition were discussed by this reviewer previously. The additional experiments using a DDAH inhibitor appear more helpful to dissect the mechanism of the diverse findings presented in this present manuscript. However, the author spent quite some time searching for the MS023 data in the revised manuscript, because the authors did not indicate in their response that this data had been deleted.

Thanks for your comments! Following the suggestion, we had performed the in vivo experiments by using a DDAH inhibitor to support our findings. Although we found that MS023 had the positive effects on bone formation, we deleted the figures and description in the present manuscript. We will go deep into it further in the future study.

7) In this criticism of the reviewer, the lack of data showing the biochemical consequences of reduced ADMA AND reeducated arginine by exercise (Figure 5) were discussed. The authors provided additional measurement results to suggest that NO production was increased by exercise. However, once again, this data is not included in the revised manuscript, so the information is not accessible for the reader. The data must be included in the revised figure or in a supplementary figure.

Thanks for your comments! This data was now included in the supplementary figures 3, and the results were discussed in the text (page 8 202-203).